# Caduceus: MoE-enhanced Foundation Models Unifying Biological and Natural Language

## Abstract

Multi-modality pre-training on protein sequences with textual descriptions has enabled general-purpose protein language models. However, as the property descriptions span heterogeneous domains, we observe a severe *data interference phenomenon*: distinct protein residues often target domain-specific annotations, revealing partially inconsistent functional mechanisms across sources, which substantially leads to degraded performance. This paper addresses this overlooked issue with a novel *Mixture of LoRA Experts (MoLE)* architecture, by efficiently fusing the knowledge across diverse property domains. Concretely, we introduce **Caduceus**, a family of MoE-enhanced foundation models built with a hierarchical pre-training paradigm to jointly integrate biological and natural language. Employing a property-guided gating router that assigns domain-specific protein tokens to different experts, the dual-granularity alignment approach reconciles signals across diverse functional mechanisms. To extend generalization beyond particular tasks, we further incorporate a multi-task instruction tuning phase, enabling robust protein parsing and natural language question answering. Extensive experiments on 17 mainstream benchmarks demonstrate that Caduceus mitigates the intrinsic data interference and consistently delivers the optimal performance. The instruction-tuned Caduceus-Instruct provides precise protein elucidation, significantly surpassing GPT-5, DeepSeek-V3, and Galactica-30B. All the models, source codes, and collected corpus will be made publicly available.

## 1 Introduction

Proteins are vital components of biological systems, playing crucial roles in catalyzing metabolic reactions and sustaining crucial biological functions (Hayes et al., 2025). Protein Language Models (PLMs) pre-trained on large-scale biological corpus have demonstrated strong capabilities to capture co-evolutionary information, driving advances in structure prediction, mutation effect estimation, and functional classification (Liu et al., 2025; Chen et al., 2025). Nevertheless, current PLMs still exhibit limited natural language understanding abilities, remaining primarily confined to property prediction tasks within specific domain, and unable to precisely interpret protein knowledge. In essence, there exists an unaddressed void in the current landscape of Large Language Models (LLMs), wherein the capacity to seamlessly traverse between biological and natural language.

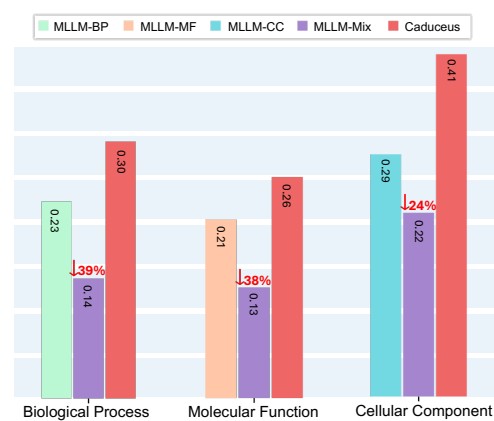

Figure 1: Mixing more training data across distinct property domains conversely leads to MLLM-Mix's performance decrease.

To bridge this gap, pioneering studies (Wang et al., 2024b; Liu et al., 2024c) delve into the construction of Multi-modal Large Language Models (MLLMs) combining protein sequences and textual annotations. While expanding annotations across diverse properties enriches textual supervision, this inevitably leads to the *data interference issue*. Concretely, protein textual annotations encompass multiple property fields, and these domain-specific properties often inconsistently correspond to

distinct protein residues across various granularity levels. Aligning such heterogeneous annotations with corresponding biological tokens inherently models diverse functional mechanisms. And treating these differing mechanisms identically induces the tug-of-war dilemma (Hadsell et al., 2020).

Herein, a simple and intuitive heuristic experiment is conducted to reveal the data interference phenomenon. As illustrated in Figure 1, we incorporated Gene Ontology annotations from three property domains (*i.e.*, Biological Process, Molecular Function, and Cellular Component) into MLLM training. Models trained independently on separate dataset exhibited strong performance on respective property prediction tasks. In contrast, joint training on mixed-domain data consistently led to an average performance decrease of 33% across all three tasks, indicating critical conflicts when integrating domain-specific property descriptions. Such interference poses a major obstacle to develop versatile foundation models unifying biological and natural language.

To mitigate such data interference issue, we propose **Caduceus**, a family of protein foundation models that leverage the Mixture of Experts (MoE) framework to integrate biological knowledge across diverse property domains. Unlike conventional MoE architectures, Caduceus employs a novel property-guided gating strategy that dynamically allocates protein tokens associated with distinct property annotations to respective experts, facilitating simultaneous accommodation of multiple domains. To preserve co-evolutionary knowledge acquired during unimodal pre-training, each expert is implemented as a Low-Rank Adaptation (LoRA) module, ensuring such fundamental information is retained. The resulting *Mixture of LoRA Experts (MoLE)* computes the optimal combination of expert weights, thereby enhancing beneficial protein characteristics while attenuating less favorable ones. Such architectural design allows distinct functional segments within a protein sequence to be governed by specialized encoding rules, uncovering their domain-specific characteristics. Motivated by this, we introduce dual-granularity training objectives to jointly capture coarse-grained and fine-grained alignment information. Eventually, the synergistic composition of the MoLE architecture and dual-granularity objectives effectively mitigates the data interference phenomenon across diverse property domains, constructing a universal bridge between biological and natural language.

Building upon the integration of multilingual knowledge, we enhance the practical value of our model by developing an advanced AI assistant capable of assembling specialized protein expertise to execute complex reasoning. To achieve general-purpose protein understanding, we incorporate instruction tuning into the hierarchical pre-training paradigm. Beyond the pre-trained protein encoder which extracts function-oriented representations, the system incorporates a cross-modal connector to bridge modality gaps and an LLM decoder that provides a textual interface grounded in profound understanding of protein properties. The instruction tuning phase endows Caduceus-Instruct with robust capabilities for interpreting protein in natural language, marking a significant step towards instruction-following foundation models capable of decoding protein scientific knowledge.

In particular, our paper makes the following contributions:

- *MoE-enhanced architecture.* We reveal pervasive data conflicts spanning diverse property domains in protein multi-modality learning. Subsequently, we develop a novel property-guided gate routing scheme within the MoLE architecture, coupled with customized dual-granularity training objectives, to mitigate the inherent data interference issue.

- *Hierarchical pre-training paradigm.* Hierarchical training framework is designed to facilitate the integration of multi-modal alignment and instruction tuning, empowering the LLM to holistically comprehend biological knowledge and conduct instruction-following Q&A.

- *Pioneering family of foundation models.* Caduceus remarkably achieves new state-of-the-art performance on 27 out of 29 downstream evaluation metrics. We hope the release of a family of MoE-enhanced protein foundation models could bring insights to the MLLM community by unifying biological and natural language.

## 2 RELATED WORK

### 2.1 MULTI-MODAL LEARNING BETWEEN PROTEIN AND LANGUAGE

Multi-modal Large Language Models (MLLMs) have continuously pushed the state-of-the-art across diverse downstream tasks through the effective incorporation of heterogeneous domain knowledge (Kim et al., 2021; Junnan et al., 2023; Liu et al., 2023; Wang et al., 2024a). Some

pioneering works integrate protein and natural language modalities effectively, leveraging extensive textual property descriptions to enhance protein representation modeling. Concretely, Liu et al. (2025); Yin et al. (2024); Dai et al. (2024); Wang et al. (2024b) present a preliminary exploration of *de novo* protein design based on natural language instructions. Additionally, researchers also develop MLLMs that specialize in elucidating proteins through annotating molecular properties and answering biological questions about specific proteins (Zhou et al., 2025b; Lv et al., 2025; Wang et al., 2025; Liu et al., 2024c). Furthermore, Zhang et al. (2022); Zhou et al. (2023); Zhang et al. (2025) performs multi-modality knowledge graph exploration to comprehensively enhance protein representation learning. Emerging research studies (Xu et al., 2023; Zhou et al., 2025a; Su et al., 2024) adopt cross-modal contrastive learning to derive function-informed protein representations, advancing abundant function prediction and bidirectional retrieval tasks.

## 2.2 MIXTURE-OF-EXPERTS

The Mixture-of-Experts (MoE) architecture is proposed to efficiently scale model parameters without correspondingly adding computational overhead (Jacobs et al., 1991; Jordan & Jacobs, 1994; Shazeer et al., 2017; Du et al., 2022). Specifically, MoE utilizes the gating router to assign distinct weights for multiple independent experts, allowing the input to flexibly activate either all experts or only a sparse combination of them. Equipped with the MoE architecture, trillion-scale large models can be trained with significantly reduced computational resources (Fedus et al., 2022; Lepikhin et al., 2021). Nevertheless, previous employments of MoE have been predominantly limited to the fields of computer vision and natural language processing (Gou et al., 2023; Chen et al., 2024a; Wu et al., 2024; Chen et al., 2024b). Our study substantially broadens the application of MoE architecture, advocating it as a compelling choice for future versatile protein language model construction.

## 2.3 PARAMETER-EFFICIENT FINE-TUNING

Recently, scaling large language models with more compute and parameters has driven significant progress in diverse natural language processing applications (Achiam et al., 2023; Guo et al., 2025; Hayes et al., 2025; Chen et al., 2025). To enable computationally efficient adaptation for specific downstream tasks, several parameter-efficient fine-tuning (PEFT) techniques have been developed (Li & Liang, 2021; Lester et al., 2021; Houlsby et al., 2019; Karimi Mahabadi et al., 2021; Hu et al., 2022). Among these, LoRA (Hu et al., 2022) stands out for its plug-and-play usability, which employs low-rank decomposition to represent weight updates via two smaller matrices, keeping the original weights frozen. In this work, we incorporate multiple LoRA experts into the MoE framework to enable efficient integration of biological and natural language.

## 3 METHOD

In Section 3.1, we first detail the MoE-enhanced architectural design to resolve the crucial tug-of-war dilemma delineated by the heuristic experiment in Figure 1. Additionally, Caduceus employs the hierarchical pre-training paradigm. In stage I, the dual-granularity integration is performed to comprehensively align protein encoder representations with natural language (Section 3.2). In stage II, the LLaVA-informed instruction tuning phase is further incorporated to endow Caduceus-Instruct with protein deciphering capabilities via instruction-following Q&A protocol. (Section 3.3).

## 3.1 MIXTURE OF LORA EXPERTS

To mitigate the critical data conflict issue, we propose the Mixture of LoRA Experts (MoLE) to facilitate protein multi-modality learning across diverse property domains. As illustrated in Figure 2, The MoLE architecture treats each layer of the trained LoRA modules as separate experts. Accordingly, it employs hierarchical weight control through a learnable gating router within each layer to learn the optimal composition weights of multiple LoRA experts. Such architectural design expands model's capacity to specialize in processing input data from broad functional fields.

The customization of expert routing mechanisms is critical for improving the performance of MoE models. Motivated by the intuition that protein functional segments belonging to distinct attribute domains operate through unique mechanisms, these tokens should be assigned to different experts for independent encoding. To accomplish this, property-guided gating router $\mathcal{G}$ is tailored, utilizing

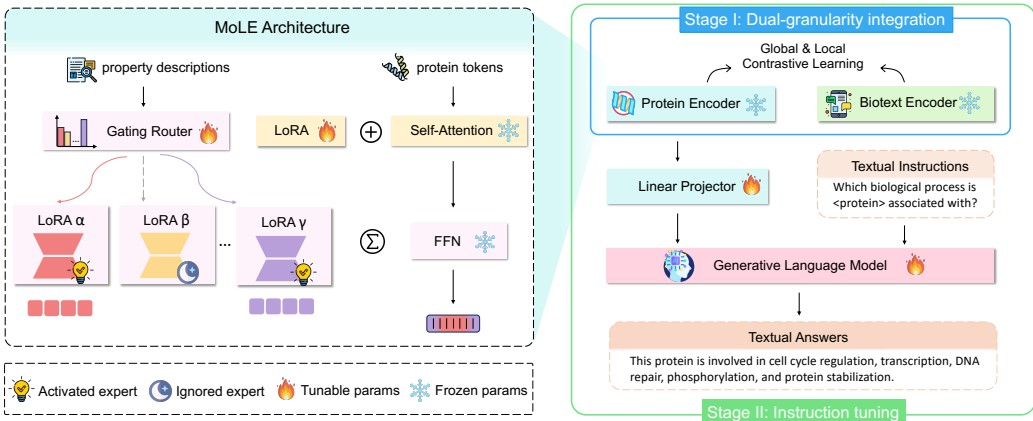

Figure 2: Overview of Caduceus hierarchical pre-training framework. Within the MoLE architecture, protein tokens associated with distinct properties are assigned to sparsely activated experts.

protein-oriented property descriptions $\mathcal{P}_{[\mathbf{x}]}$ as input to predict routing scores for protein tokens. $\epsilon$ adds stochasticity into the routing process, and $\tau$ is the temperature hyperparameter.

$$\mathcal{G}(x) = \text{Softmax}(top_k(\left(W_g \mathcal{P}_{[\mathbf{x}]} + \epsilon\right)/\tau), \tag{1}$$

To empirically balance domain-specific mechanisms, we introduce a group of independent LoRA experts within each Feed-Forward Network (FFN). The incorporated LoRA experts are sparsely activated based on calculated routing scores, while the rest are simply ignored for current protein instance. Subsequently, LoRA efficiently learns decomposed parameter updates through the product of low-rank matrices $A_i$ and $B_i$.

$$f'_{\text{FFN}}(x) = f_{\text{FFN}}(x) + \sum_{i=1}^{k} \mathcal{G}_i(x)E_i(x) = f_{\text{FFN}}(x) + \sum_{i=1}^{k} \mathcal{G}_i(x)\frac{\alpha}{r}B_i A_i x. \tag{2}$$

Previous studies (Chen et al., 2024b; Wu et al., 2024; Mu & Lin, 2025) have elucidated the imbalance phenomenon where the gating router tends to converge to a specific subset of high-performing LoRAs early in the training process. Distinctive characteristics of the remaining LoRA experts are primarily diminished or lost, thus impeding the model from synthesizing diverse domain-specific knowledge. To alleviate this, we incorporate a load-balancing loss to encourage equal activation across all experts. $c_j$ and $p_j$ denote the assigned token numbers and total routing probability for the $j$-th expert, respectively. The auxiliary loss is minimized when the dispatching is perfectly balanced.

$$\mathcal{L}_{\text{LB}} = \sum_{j=1}^{K} c_j p_j = \sum_{j=1}^{K} c_j \cdot \sum_{x \in X} \frac{\exp(\mathcal{G}_j(x))}{\sum_j \exp(\mathcal{G}_j(x))}. \tag{3}$$

## 3.2 DUAL-GRANULARITY INTEGRATION

### 3.2.1 GLOBAL ALIGNMENT

The dual-granularity integration stage utilizes ESM-2 (Lin et al., 2023) and BiomedBERT (Gu et al., 2021) to encode protein sequences and biomedical text, respectively. Global contrastive loss (Radford et al., 2021) is employed to achieve coarse-grained alignment between the multi-modal feature spaces of proteins and biotexts $(S, T)$. More precisely, we attain global alignment by maximizing the similarity to the corresponding embeddings in the other modality (protein-to-biotext and biotext-to-protein), while minimizing the similarity to non-matching embeddings within the batch.

$$\mathcal{L}_{\text{GC}} = -\frac{1}{2} \left[ \mathbb{E}_{p(S,T)}(log \frac{\exp(\phi(S_i, T_i)/\tau_1)}{\sum_j \exp(\phi(S_i, T_j)/\tau_1)}) + \mathbb{E}_{p(T,S)}(log \frac{\exp(\phi(T_i, S_i)/\tau_1)}{\sum_j \exp(\phi(T_i, S_j)/\tau_1)}) \right]. \tag{4}$$

where $\phi(S_i, T_j) = \frac{S_i}{\|S_i\|_2} \cdot \frac{T_j}{\|T_j\|_2}$ and $\tau_1$ denotes the temperature controlling the softmax distribution.

### 3.2.2 LOCAL ALIGNMENT

As usually specific biological properties correspond to contiguous or discrete protein functional segments, we additionally incorporate local alignment losses (*i.e.*, biotext-guided static segment reconstruction and property-grouped dynamic segment alignment) to facilitate the injection of fine-grained information (Zhou et al., 2025a). Specifically, *Biotext-guided Static segment Reconstruction (BSR)* leverages information from both modalities to reconstruct corrupted static segments. A spectrum of sampling iterations are conducted to construct a random collection of non-overlapping static segments for subsequent masking and reconstruction. The combined length of these segments constitutes 15% of the total length of protein sequences. Furthermore, a cross-modality reconstruction module is introduced to achieve feature fusion and segment generation. Concretely, we employ a cross-attention block to facilitate deep fusion between protein and biotext features, while an MLP is utilized to predict the reconstructed tokens at masked positions.

$$\mathcal{L}_{\text{BSR}} = \mathbb{E}_{p(T,e^m)}\text{CrossEntropy}(\Phi(T, e^m), y_e). \tag{5}$$

where $\Phi(T, e^m)$ is the predicted probability of protein sequence with masked static segments $e^m$, and $y_e$ is the corresponding ground truth.

*Property-grouped Dynamic segment Alignment (PDA)* improves alignment precision between property-grouped dynamic segments and corresponding descriptive attributes. Concretely, a prototype memory bank is developed to capture the semantic essence of biological property descriptions, omitting the precise retention of redundant syntactic details. Given the aggregated property prototypes $a_i$, we sparsify and min-max normalize the similarity matrix between property prototypes and protein sequence tokens. The selected tokens exhibiting similarity scores exceeding the predefined threshold constitute property-grouped dynamic segments $e_i$. We innovatively optimizes for the fine-grained alignment between aggregated property prototypes and their respective property-grouped dynamic segments, alleviating the inter-domain interference across multiple attribute spaces.

$$\mathcal{L}_{\text{PDA}} = -\frac{1}{2}\left[\mathbb{E}_{p(a,e)}(log\frac{\exp(\phi(a_i, e_i)/\tau_2)}{\sum_k \exp(\phi(a_i, e_k)/\tau_2)}) + \mathbb{E}_{p(e,a)}(log\frac{\exp(\phi(e_i, a_i)/\tau_2)}{\sum_k \exp(\phi(e_i, a_k)/\tau_2)})\right]. \tag{6}$$

where $\phi(e_i, a_k) = \frac{e_i}{\|e_i\|_2} \cdot \frac{a_k}{\|a_k\|_2}$ and $\tau_2$ is the temperature parameter that modulates the softmax.

### 3.3 INSTRUCTION TUNING

The multi-modal alignment pre-training stage introduced in Section 3.2 empowers the model an extensive comprehension capability of both natural language and protein language. Nevertheless, similar to most existing models, the current model still heavily relies on task-specific supervised fine-tuning, thereby limiting its generalization to predicting only specific functional properties (Xu et al., 2023; Duan et al., 2025). To address this challenge, we further incorporate the multi-task instruction tuning stage to enhance general-purpose textual protein understanding through biological knowledge-based Q&A protocol. Concretely, informed by LLaVA architecture, the multi-modal aligned protein encoder acts as the biological assistant. We additionally introduce a cross-modal connector and employ LLaMA3 (Dubey et al., 2024) as the natural language decoder. Instruction tuning effectively bridges the gap between the next-word prediction objective of LLMs and the user's goal of obtaining responses that adhere closely to human instructions (Ouyang et al., 2022; Liu et al., 2024b). Formally, a multi-modal instruction sample can be represented as a triplet form $(\mathcal{I}, \mathcal{M}, \mathcal{R})$, where $\mathcal{I}$ denotes the user instruction, $\mathcal{M}$ corresponds to the uploaded protein, and $\mathcal{R}$ refers to the ground-truth response. The MLLM is prompted to parse user instructions, and generate elaborate analyses corresponding to the encoded function-oriented protein representations.

$$\mathcal{L}_{\text{Instruct}} = -\sum_i \log p_\theta(\mathcal{R}_i \mid \mathcal{I}, \mathcal{M}, \mathcal{R}_{<i}). \tag{7}$$

## 4 EXPERIMENTS

### 4.1 EXPERIMENTAL SETTING

The full-fledged Caduceus undergoes a hierarchical pre-training paradigm, consisting of the dual-granularity integration stage aligns protein and biotext feature spaces (Section 3.2), followed by

the instruction tuning stage steers LLM towards textual interpretation of protein attributes (Section 3.3). To achieve this, we collect a diverse pre-training corpus from UniProtKB (Consortium, 2019), RCSB-PDB (Berman et al., 2000), Enzyme Commission (Bairoch, 2000), and Gene Ontology (Ashburner et al., 2000) databases. 251 million proteins are acquired, each accompanied by factual functional annotations covering expansive property fields. Abundant instruction templates are manually designed, and GLM-4 (GLM et al., 2024) is prompted to transform filtered high-confidence annotations into natural language Q&A samples. The resulting instruction dataset contains 5 million logical reasoning instances, comprising open-ended and closed-set Q&A paradigms. For model deployment, the protein and biotext encoders are implemented with ESM-2-650M and BiomedBERT-100M. A simple linear projection layer is utilized as the cross-modal connector to reduce information loss. And we keep LLaMA3-8B (Dubey et al., 2024) as Caduceus-Instruct's final LLM decoder after elaborate ablation analysis (Section 4.6) . We build our codes upon the PyTorch framework and utilize 64 Tesla V100 GPUs with 2048 batch size. An Adam optimizer with $1e^{-5}$ learning rate is utilized for training, consuming a total of 12,000 GPU hours. Further implement details are described in Appendix A.4.

## 4.2 CADUCEUS ACCURATELY PREDICTS PROTEIN FUNCTIONAL PROPERTIES

Accurate protein functional characterization holds paramount significance in the realms of biology and biomedicine, enabling researchers to precisely identify and target specific proteins involved in critical disease pathways. We first conduct protein functional property prediction experiments to validate the efficacy of multi-modality alignment between biological language and natural language. Overall, 11 benchmark tasks across three task types, including localization classification, mutation effect prediction, and biological function annotation, are integrated for assessment (see thorough task description in Appendix A.5). We adhere to the official standard data splits for all evaluation tasks following Xu et al. (2023). To facilitate comprehensive comparison, we incorporate a diverse spectrum of baselines, including four traditional protein models trained from scratch (*i.e.*, CNN, ResNet, LSTM, and Transformer) as well as five protein large language models (*i.e.*, ProtBERT (Elnaggar et al., 2022), OntoProtein (Zhang et al., 2022), ESM-1b (Rives et al., 2021), ESM-2 (Lin et al., 2023), and ProtST (Xu et al., 2023)). Diverse assessment criteria are employed, including Acc, Spearman's $\rho$, AUPR, and $F_{max}$. To enhance the credibility of our experiments, we simultaneously report the evaluation results under both linear probing and full fine-tuning configurations.

Table 1: Evaluation results on 11 mainstream benchmarks, including Loc class, Effect pred, and Function anno. We comprehensively report evaluation results for linear probing and full fine-tuning.

| Models | Loc class (Acc%) | | Effect pred (Spearman's $\rho$) | | | | | Function anno (AUPR & $F_{max}$) | | | | | | | |
|---|---|---|---|---|---|---|---|---|---|---|---|---|---|---|---|
| | Bin | Sub | $\beta$-lac | AAV | Thermo | Flu | Sta | EC | | GO-BP | | GO-MF | | GO-CC | |
| *Traditional model performance trained from scratch* | | | | | | | | | | | | | | | |
| CNN | 82.67 | 58.73 | 0.781 | 0.746 | 0.494 | 0.682 | 0.637 | 0.540 | 0.545 | 0.165 | 0.244 | 0.380 | 0.354 | 0.261 | 0.387 |
| ResNet | 78.99 | 52.30 | 0.152 | 0.739 | 0.528 | 0.636 | 0.126 | 0.137 | 0.187 | 0.166 | 0.280 | 0.281 | 0.267 | 0.266 | 0.403 |
| LSTM | 88.11 | 62.98 | 0.139 | 0.125 | 0.564 | 0.494 | 0.533 | 0.032 | 0.082 | 0.130 | 0.248 | 0.100 | 0.166 | 0.150 | 0.320 |
| Transformer | 75.74 | 56.02 | 0.261 | 0.681 | 0.545 | 0.643 | 0.649 | 0.187 | 0.219 | 0.135 | 0.257 | 0.172 | 0.240 | 0.170 | 0.380 |
| *LLM performance under linear probing* | | | | | | | | | | | | | | | |
| ProtBERT | 81.54 | 59.44 | 0.616 | 0.209 | 0.562 | 0.339 | 0.697 | 0.028 | 0.089 | 0.130 | 0.245 | 0.053 | 0.120 | 0.143 | 0.296 |
| OntoProtein | 84.87 | 68.34 | 0.471 | 0.217 | 0.605 | 0.432 | 0.688 | 0.411 | 0.417 | 0.243 | 0.345 | 0.418 | 0.383 | 0.346 | 0.465 |
| ESM-1b | 91.61 | 79.82 | 0.528 | 0.454 | 0.674 | 0.430 | 0.750 | 0.649 | 0.642 | 0.309 | 0.403 | 0.557 | 0.528 | 0.404 | 0.504 |
| ESM-2 | 91.32 | 80.84 | 0.559 | 0.374 | 0.677 | 0.456 | 0.746 | 0.711 | 0.694 | 0.311 | 0.412 | 0.577 | 0.547 | 0.404 | 0.519 |
| ProtST | 92.52 | 83.39 | 0.565 | 0.398 | 0.681 | 0.499 | 0.776 | 0.810 | 0.784 | 0.358 | 0.458 | 0.643 | 0.601 | 0.451 | 0.546 |
| **Caduceus-650M** | 94.39 | 83.65 | 0.565 | 0.532 | 0.682 | 0.503 | 0.795 | 0.827 | 0.808 | 0.453 | 0.556 | 0.635 | 0.598 | 0.492 | 0.586 |
| **Caduceus-3B** | *95.48* | 85.11 | 0.637 | 0.602 | *0.689* | 0.585 | 0.810 | 0.871 | 0.859 | 0.462 | 0.567 | 0.673 | 0.656 | *0.535* | *0.608* |
| *LLM performance under full fine-tuning* | | | | | | | | | | | | | | | |
| ProtBERT | 91.32 | 76.53 | 0.731 | 0.794 | 0.660 | 0.679 | 0.771 | 0.859 | 0.838 | 0.188 | 0.279 | 0.464 | 0.456 | 0.234 | 0.408 |
| OntoProtein | 92.47 | 77.59 | 0.757 | 0.791 | 0.662 | 0.630 | 0.731 | 0.854 | 0.841 | 0.284 | 0.436 | 0.603 | 0.631 | 0.300 | 0.441 |
| ESM-1b | 92.40 | 78.13 | 0.839 | 0.821 | 0.669 | 0.679 | 0.694 | 0.884 | 0.869 | 0.332 | 0.452 | 0.630 | 0.659 | 0.324 | 0.477 |
| ESM-2 | 91.72 | 78.67 | 0.867 | 0.817 | 0.672 | 0.677 | 0.718 | 0.888 | 0.874 | 0.340 | 0.472 | 0.643 | 0.662 | 0.350 | 0.472 |
| ProtST | 92.52 | 80.22 | 0.879 | 0.825 | 0.682 | 0.682 | 0.738 | 0.898 | 0.878 | 0.342 | 0.482 | 0.647 | 0.668 | 0.364 | 0.487 |
| **Caduceus-650M** | 95.08 | *85.34* | *0.884* | *0.892* | 0.686 | *0.685* | *0.819* | *0.906* | *0.898* | *0.467* | *0.574* | *0.687* | *0.696* | 0.515 | 0.592 |
| **Caduceus-3B** | **96.01** | **87.26** | **0.893** | **0.915** | **0.693** | **0.687** | **0.830** | **0.910** | **0.908** | **0.512** | **0.577** | **0.698** | **0.704** | **0.596** | **0.613** |

As displayed in Table 1, we observe that multi-modal aligned PLMs clearly outperform the vanilla PLMs, demonstrating the benefits of injecting textual knowledge into PLM pre-training process. Notably, although traditional models (*e.g.*, CNN) deliver strong competitors in mutation effect pre-

diction tasks, Caduceus still sustains its leading performance, achieving improvements of 30% and 40% on the Sta and Thermo benchmarks, respectively. Overall, Caduceus establishes state-of-the-art results on all 11 mainstream functional property prediction benchmarks.

## 4.3 CADUCEUS EFFICIENTLY EXECUTES CROSS-MODAL RETRIEVAL

The cross-modal retrieval accomplishes the translation of complex protein landscapes into comprehensible language descriptions, as well as the identification of proteins corresponding to particular biological characteristics. Here we evaluate the cross-modal retrieval capability on the ProteinKG25 (Zhang et al., 2022), a large-scale knowledge graph database containing numerous protein sequences aligned with biology knowledge facts. The triples of the same protein are aggregated to construct corresponding textual descriptions. Bidirectional cross-modal retrieval tasks are conducted using the data split of 422,315/10,000/10,000. To highlight the technical superiority of proposed method, various multi-modal aligned LLMs are incorporated as evaluation baselines, including ProtST (Xu et al., 2023), ProteinCLAP (Liu et al., 2025), and ProtT3 (Liu et al., 2024c). We utilize Accuracy and Recall@20 as the evaluation metrics to assess the precision of protein-to-text and text-to-protein retrieval. Notably, multiple searching settings, including both single-batch and entire-database retrieval, are applied to further enhance the experimental credibility.

Table 2: Cross-modal retrieval performance under the single-batch searching regime.

| Models | Protein-to-Text | | Text-to-Protein | |
|---|---|---|---|---|
| | Accuracy | Recall@20 | Accuracy | Recall@20 |
| ProtST | 70.8 | 98.5 | 70.9 | 98.2 |
| ProteinCLAP | 93.2 | 99.2 | 93.2 | 99.3 |
| ProtT3 | 95.1 | **99.9** | 95.3 | **99.9** |
| **Caduceus** | **97.2** | **99.9** | **98.6** | **99.9** |

Table 3: Cross-modal retrieval performance under the entire-database searching regime.

| Models | Protein-to-Text | | Text-to-Protein | |
|---|---|---|---|---|
| | Accuracy | Recall@20 | Accuracy | Recall@20 |
| ProtST | 5.5 | 41.6 | 5.8 | 43.3 |
| ProteinCLAP | 39.0 | 89.4 | 39.3 | 89.7 |
| ProtT3 | 55.8 | 91.7 | 55.6 | 91.7 |
| **Caduceus** | **62.1** | **97.5** | **63.6** | **97.9** |

As presented in Table 2 and Table 3, we observe that Caduceus outperforms ProtT3 by 6% and 8% accuracy within the entire retrieval database, highlighting its capability in aligning proteins with corresponding textual descriptions. Collectively, Caduceus achieves optimal performance across all baselines on 6 out of 8 evaluation metrics, while matching ProtT3 results on the remaining 2 metrics.

Moreover, we also present a qualitative application to demonstrate Caduceus can effectively retrieve physiologically relevant immune proteins. Concretely, we download all 1,030,081 protein sequences from the RCSB-PDB (Protein Data Bank) archive (Berman et al., 2000). Caduceus is then queried to identify SARS-CoV-2 antibodies with potential clinical significance. An exhaustive all-versus-all search is performed within the sorted protein corpus based on the representation similarities. Experimentally-determined 3D structures of the top-ranked antibodies with corresponding antigens are depicted in Figure 3. We concurrently provide the protein attribute annotations from the PDB database. The identified stable and regular antigen-antibody structures underscore Caduceus's significant potential to accelerate the controllable protein discovery in real-world scenarios.

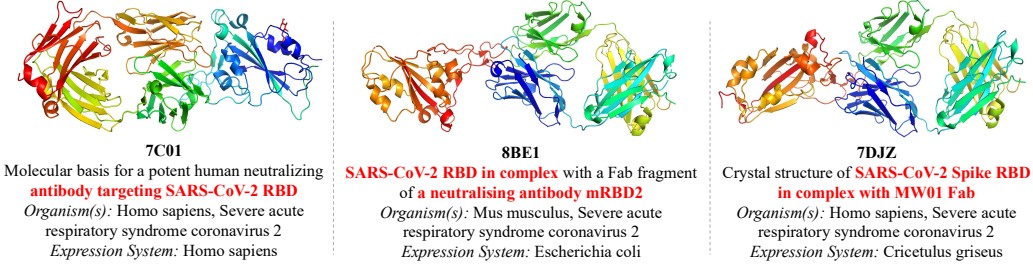

**7C01**
Molecular basis for a potent human neutralizing **antibody targeting SARS-CoV-2 RBD**
*Organism(s):* Homo sapiens, Severe acute respiratory syndrome coronavirus 2
*Expression System:* Homo sapiens

**8BE1**
**SARS-CoV-2 RBD in complex** with a Fab fragment of **a neutralising antibody mRBD2**
*Organism(s):* Mus musculus, Severe acute respiratory syndrome coronavirus 2
*Expression System:* Escherichia coli

**7DJZ**
Crystal structure of **SARS-CoV-2 Spike RBD in complex with MW01 Fab**
*Organism(s):* Homo sapiens, Severe acute respiratory syndrome coronavirus 2
*Expression System:* Cricetulus griseus

Figure 3: 3D structure visualization of the top-ranked retrieved candidates. (Best viewed in color)

## 4.4 CADUCEUS FACILITATES THE TEXTUAL INTERPRETATION OF PROTEINS

Harnessing LLMs to drive advancements in textual protein understanding is essential for elucidating cellular mechanisms (Yin et al., 2025). Herein, we demonstrate that Caduceus-Instruct enables the textual interpretation of proteins through diverse biological knowledge-based Q&A paradigms.

Table 4: Evaluation performance on text-based protein understanding experiment. The evaluation results for distinct Q&A settings (*i.e.*, open-ended generation and closed-set answer) are reported.

| Models | Open-Ended | | | | | | | | | Closed-Set |
|---|---|---|---|---|---|---|---|---|---|---|
| | BLEU-2 | BLEU-4 | ROUGE-1 | ROUGE-2 | ROUGE-L | METEOR | BERT-P | BERT-R | BERT-F1 | Accuracy |
| *LLM performance in zero-shot* | | | | | | | | | | |
| GPT-4o | 3.26 | 0.14 | 15.02 | 3.73 | 12.30 | 13.01 | 85.28 | 86.51 | 85.86 | 59.70% |
| DeepSeek-V3 | 2.75 | 0.02 | 12.15 | 3.37 | 10.54 | 10.25 | 85.32 | 85.63 | 85.44 | 56.49% |
| Galactica | 0.43 | 0.01 | 3.49 | 0.41 | 2.67 | 2.44 | 85.79 | 82.61 | 84.08 | 39.15% |
| BioT5+ | 3.88 | 1.92 | 12.12 | 4.88 | 10.37 | 14.26 | 85.14 | 85.93 | 85.48 | — |
| InstructProtein | 5.50 | 2.97 | 14.80 | 5.68 | 13.76 | 13.17 | 85.34 | 85.92 | 85.57 | 48.37% |
| *LLM performance through instruction-tuning* | | | | | | | | | | |
| OpenLLaMA | 36.19 | 30.65 | 48.33 | 36.52 | 45.53 | 49.01 | 92.92 | 91.87 | 92.35 | 71.77% |
| LLaMA2-Chat | 57.02 | 49.47 | 70.80 | 57.24 | 67.78 | 65.96 | 94.95 | 95.17 | 95.05 | 71.68% |
| ProtT3 | 58.58 | 50.27 | 70.28 | 56.27 | 67.93 | 66.38 | 94.92 | 95.28 | 95.51 | 77.98% |
| SEPIT | 58.43 | 51.04 | 72.34 | 58.77 | 69.13 | 67.91 | 95.32 | 95.59 | 95.44 | 79.05% |
| **Caduceus-w/o-MoLE** | 56.49 | 50.28 | 71.75 | 58.00 | 68.92 | 65.15 | 95.19 | 95.25 | 95.37 | 72.04% |
| **Caduceus-650M-Instruct** | *62.74* | *56.39* | *76.86* | *63.01* | *74.45* | *70.13* | **95.92** | *96.00* | *95.87* | *80.66%* |
| **Caduceus-3B-Instruct** | **68.48** | **62.09** | **81.23** | **69.05** | **79.94** | **73.75** | *95.77* | **96.59** | **96.48** | **83.84%** |

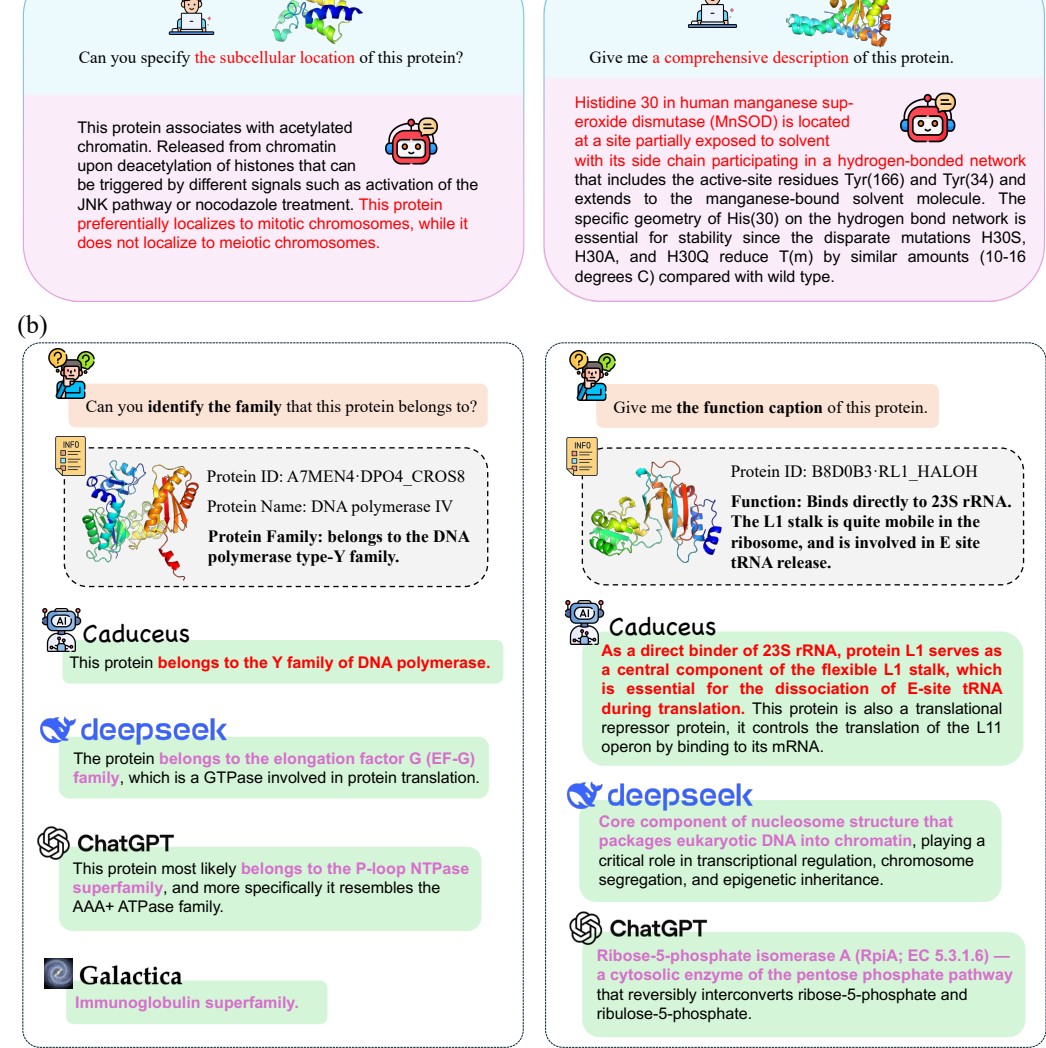

Figure 4: Qualitative results of utilizing LLMs to facilitate the textual interpretation of proteins. We incorporate DeepSeek-V3, GPT-5, and Galactica-30B for comparative analysis. LLM responses **matching** and mismatching the fundamental knowledge from protein ID cards are highlighted.

Specifically, the open-ended generation task, comprising 14,411 instruction instances, requires the model to provide a detailed elucidation regarding the biological properties of proteins. In contrast, the closed-set answer task, consisting of 61,566 instruction instances, is designed to perform correctness discrimination in the context of specific biological questions.

To facilitate comprehensive comparison, we include current mainstream general-purpose LLMs providing API services (*i.e.*, GPT-4o (Achiam et al., 2023), and DeepSeek-V3 (Liu et al., 2024a)), as well as open-source LLMs tailored for injecting biological knowledge (*i.e.*, Galactica (Taylor et al., 2022), BioT5+ (Pei et al., 2024), and InstructProtein (Wang et al., 2024b)), to perform zero-shot inference. Diverse LLMs utilizing instruction tuning strategy (*i.e.*, OpenLLaMA (Geng & Liu, 2023), LLaMA2-Chat (Touvron et al., 2023), and SEPIT (Wu et al., 2025)) are also incorporated for comparison. We employ BLEU, ROUGE, METEOR, BERTScore, and Accuracy as evaluation metrics.

As shown in Table 4, diverse LLMs operating under the zero-shot paradigm exhibit limited performance, persisting across powerful general-purpose models GPT-4o and DeepSeek-V3. Additionally, open-source LLMs fine-tuned on biomedical corpora exhibit decent enhancement, highlighting the necessity of multi-modal information fusion. Excitingly, Caduceus-Instruct consistently outperforms all baseline models by a considerable margin, further verifying the incorporated instruction-tuning phase endows our model with the capability of precise protein textual interpretation.

Herein, we also provide complementary qualitative analyses to underscore practical utility of the constructed model. In Figure 4(a), we first reveal that Caduceus-Instruct is capable of providing targeted and thorough responses to user queries regarding specific biological properties. Furthermore, Figure 4(b) validates the superiority of Caduceus-Instruct over leading-edge LLMs (*i.e.*, DeepSeek-V3 (Liu et al., 2024a), GPT-5 (OpenAI, 2025), and Galactica-30B (Taylor et al., 2022)) in accurately parsing protein properties. Surprisingly, Caduceus-Instruct delivers precise responses to diverse user questions, whereas other LLMs incorporate incorrect information to varying degrees, exhibiting the hallucination behavior. This demonstrates Caduceus can function as a versatile AI assistant to facilitate textual deciphering of proteins.

## 4.5 CADUCEUS PRECISELY MODELS PROTEIN INTERACTION MECHANISM

For local protein interaction mechanism, amino acid contact prediction aims to predict whether two protein tokens within the same sequence are in contact, which is a token-level binary classification task. Specifically, we adopt the TAPE benchmark (Rao et al., 2019), with LSTM, ResNet, Transformer, ProtBERT, ESM-1b, and ESM-2 serving as baselines. Importantly, we incorporate recent excellent protein multi-modal models to facilitate comprehensive evaluation, including OntoProtein (Zhang et al., 2022), KeAP (Zhou et al., 2023), Kara (Zhang et al., 2025). Precision scores P@L, P@L/5, and P@L/2 are defined as the precision evaluated at the top L, L/2, and L/5 predictions, respectively. For global protein interaction mechanism, Protein-Protein Interaction (PPI) identification aims to predict the interaction state of two protein sequences, which is a sequence-level multi-label classification task. We incorporate SHS27K, SHS148K, and STRING as PPI evaluation benchmarks (Chen et al., 2019; Lv et al., 2021). Diverse GNNs (*i.e.*, DPPI, DNNPPI, PIPR, GNN-PPI), PLMs (*i.e.*, ProtBERT, ESM-1b, ESM-2), and knowledge-exploited protein encoders (*i.e.*, OntoProtein, KeAP, Kara) are established baselines. F1 score is utilized as the assessment metric.

Table 5: Experimental results of amino acid contact prediction. *seq* means the number of amino acids separating two selected protein tokens.

| Models | 6 ≤ seq ≤ 12 | | | 12 ≤ seq ≤ 24 | | | 24 ≤ seq | | |
|---|---|---|---|---|---|---|---|---|---|
| | P@L | P@L/2 | P@L/5 | P@L | P@L/2 | P@L/5 | P@L | P@L/2 | P@L/5 |
| LSTM | 0.26 | 0.36 | 0.49 | 0.20 | 0.26 | 0.34 | 0.20 | 0.23 | 0.27 |
| ResNet | 0.25 | 0.34 | 0.46 | 0.28 | 0.25 | 0.35 | 0.10 | 0.13 | 0.17 |
| Transformer | 0.28 | 0.35 | 0.46 | 0.19 | 0.25 | 0.33 | 0.17 | 0.20 | 0.24 |
| ProtBERT | 0.30 | 0.40 | 0.52 | 0.27 | 0.35 | 0.47 | 0.20 | 0.26 | 0.34 |
| ESM-1b | 0.38 | 0.48 | 0.62 | 0.33 | 0.43 | 0.56 | 0.26 | 0.34 | 0.45 |
| ESM-2 | 0.40 | 0.50 | 0.62 | 0.35 | 0.44 | 0.56 | 0.27 | 0.35 | 0.45 |
| OntoProtein | 0.37 | 0.46 | 0.57 | 0.32 | 0.40 | 0.50 | 0.24 | 0.31 | 0.39 |
| KeAP | 0.41 | 0.51 | 0.63 | 0.36 | 0.45 | 0.54 | 0.28 | 0.35 | 0.43 |
| Kara | 0.45 | 0.55 | 0.65 | 0.39 | 0.48 | 0.59 | *0.31* | *0.39* | 0.48 |
| **Caduceus-650M** | *0.47* | *0.58* | *0.67* | *0.43* | *0.51* | *0.61* | 0.30 | 0.36 | *0.50* |
| **Caduceus-3B** | **0.50** | **0.62** | **0.69** | **0.44** | **0.55** | **0.63** | **0.35** | **0.47** | **0.53** |

Table 6: Evaluation performance of protein protein interaction prediction using BFS and DFS.

| Models | SHS27K | | | SHS148K | | | STRING | | |
|---|---|---|---|---|---|---|---|---|---|
| | BFS | DFS | Avg | BFS | DFS | Avg | BFS | DFS | Avg |
| DNN-PPI | 48.09 | 54.34 | 51.22 | 57.40 | 58.42 | 57.91 | 53.05 | 64.94 | 59.00 |
| DPPI | 41.43 | 46.12 | 43.77 | 52.12 | 52.03 | 52.08 | 56.68 | 66.82 | 61.75 |
| PIPR | 44.48 | 57.80 | 51.14 | 61.83 | 63.98 | 62.91 | 55.65 | 67.45 | 61.55 |
| GNN-PPI | 63.81 | 74.72 | 69.27 | 71.37 | 82.67 | 77.02 | 78.37 | 91.07 | 84.72 |
| ProtBERT | 70.94 | 73.36 | 72.15 | 70.32 | 78.86 | 74.59 | 67.61 | 87.44 | 77.53 |
| ESM-1b | 74.92 | 78.83 | 76.88 | 77.49 | 82.13 | 79.31 | 78.54 | 88.59 | 83.57 |
| ESM-2 | 75.05 | 79.55 | 77.30 | 77.19 | 83.34 | 80.26 | 81.32 | 89.19 | 85.30 |
| OntoProtein | 72.26 | 78.89 | 75.58 | 75.23 | 77.52 | 76.38 | 76.71 | 91.45 | 84.08 |
| KeAP | 78.58 | 77.54 | 78.06 | 77.22 | 84.74 | 80.98 | 81.44 | 89.77 | 85.61 |
| Kara | 81.18 | 78.85 | 80.01 | 79.62 | 86.02 | 82.82 | *82.73* | 92.46 | *87.59* |
| **Caduceus-650M** | *82.13* | *79.26* | *80.69* | *82.25* | *87.63* | *84.94* | 81.25 | **92.84** | 87.04 |
| **Caduceus-3B** | **84.51** | **80.25** | **82.38** | **83.64** | **89.52** | **86.58** | **83.27** | *92.63* | **87.95** |

As shown in Table 5 and Table 6, Kara demonstrates promising evaluation performance, particularly in challenging long-range amino acid contact predictions and the STRING PPI identification. In contrast, Caduceus achieves new state-of-the-art performance on all 18 evaluation metrics, surpassing previous excellent protein multi-modal baselines. In most cases, Caduceus-3B exhibits significantly superior performance compared to Caduceus-650M, providing further evidence for the efficacy of parameter scaling. Leveraging the MoLE architecture to mitigate data conflict across distinct properties, our model unlocks comprehensive and precise protein interaction mechanism modeling.

### 4.6 ABLATION STUDIES

**Mixture of LoRA Experts.** To study the impact of MoE-enhanced architectural design, we compare Caduceus's sparsely activated MoLE with the plain-LoRA approach. In Table 7, Caduceus achieves holistic performance superiority over the plain-LoRA approach, indicating that MoLE effectively mitigates data interference across diverse property domains. Furthermore, we present the inference workload distribution of distinct experts in Figure 5. The average activation ratios across different layers are investigated to be 15.9%, 11.1%, 16.4%, 16.2%, 20.4%, and 20.0%. The well-balanced activation of LoRA experts demonstrates that the load-balancing loss facilitates the parallel encoding mechanism.

| Models | Sub | Thermo | EC | |
|---|---|---|---|---|
| | Acc% | Spearman's $\rho$ | AUPR | $F_{\max}$ |
| Caduceus | **83.65** | **0.682** | **0.827** | **0.808** |
| plain-LoRA | 78.32 | 0.665 | 0.687 | 0.685 |
| w/o $\mathcal{L}_{\text{BSR}}$ | 83.01 | 0.680 | 0.799 | 0.773 |
| w/o $\mathcal{L}_{\text{PDA}}$ | 81.54 | 0.678 | 0.721 | 0.694 |

Table 7: Ablation study of the MoLE architecture and training objectives.

**Dual-Granularity Objectives.** Besides classical CLIP contrastive loss (Radford et al., 2021), we additionally incorporate BSR and PDA to combine dual-granularity training objectives. The ablation results with full or partial pre-training objectives are reported in Table 7. The metrics indicate that both BSR and PDA are essential for injecting fine-grained information, and the absence of PDA causes a more substantial performance degradation compared to that of BSR.

**Generative LLM Selection.** We ablate the choice of generative LLMs for text-based protein understanding. Figure 6 displays comparative results for open-ended protein Q&A. Deploying various LLMs decoders (*i.e.*, Galactica-6.7B (Taylor et al., 2022), Vicuna-13B (Chiang et al., 2023), LLaMA3-8B (Dubey et al., 2024)), identical instruction tuning phases are carried out to elucidate diverse biological properties of proteins. The performance disparities indicate that LLaMA3 is clearly a superior option than the other two LLMs for deployment in protein textual interpretation.

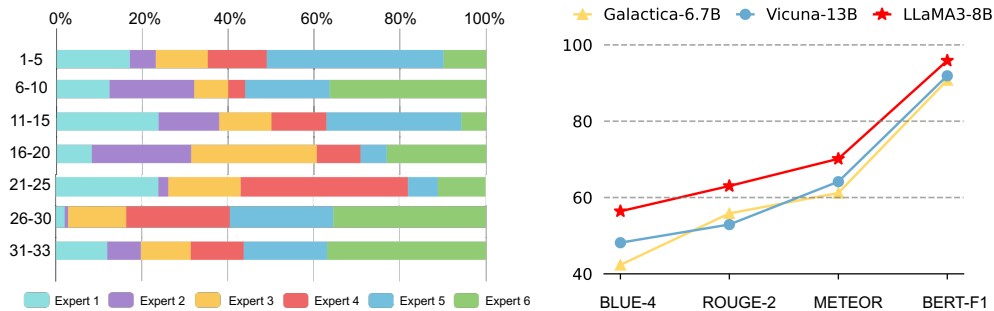

Figure 5: MoLE Loading percentage visualization. Figure 6: Q&A employing distinct LLM decoders.

## 5 CONCLUSION

In this paper, we incorporate the Mixture of LoRA Experts (MoLE) to effectively mitigate the intrinsic data interference across diverse property domains. Building on this, we train Caduceus, a series of foundation models to unify biological and natural language. Dual-granularity integration stage fuses multi-modal information to derive function-centric protein representation. Instruction tuning phase further endows Caduceus-Instruct with the capability of biological knowledge-based Q&A interactions. Through extensive evaluation, Caduceus achieves new state-of-the-art performance on 17 challenging benchmarks, exhibiting immense application prospects in accurate function prediction, efficient cross-modal retrieval, and nuanced textual interpretation of proteins. This work is expected to stimulate future research on efficiently developing versatile multi-modality foundation models.

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

## A  APPENDIX

### A.1  LARGE LANGUAGE MODEL USAGE

During the research work, we employ large language models solely as automated proofreading agents to polish the academic English expressions in this paper. Specifically, the innovative motivation of this research work is independently conceived by the human author team, without involvement from the large language models. Furthermore, the entire manuscript is also originally written by the human authors. The large language models are only utilized in the final stage to assist in refining key sections of the English expressions and word choices.

### A.2  ETHICS STATEMENT

This study incorporates biological knowledge-based Q&A, which is built upon publicly available pre-trained large language model Galactica-6.7B (Taylor et al., 2022), Vicuna-13B (Chiang et al., 2023), and LLaMA3-8B (Dubey et al., 2024). Moreover, this study is solely confined to *in silico* validation on existing protein datasets and does not involve any wet-laboratory experimental procedures. As such, this work does not introduce any additional ethical concerns beyond those already associated with the use of open-source large language models and standard benchmarks. All experimental procedures are conducted in strict compliance with the ethical guidelines and terms of use governing both the models and datasets employed in this study.

### A.3  REPRODUCIBILITY STATEMENT

We have thoroughly elucidated our architectural design, training paradigm, and implementation details throughout the paper to facilitate the comprehensive and precise understanding. Concretely, we elaborate on the architectural design and training strategy in Section 3.1 and 3.2, respectively. We implement the source code with PyTorch, and the detailed training settings are illustrated in Section 4.1. Moreover, detailed explanations of the utilized test datasets and inference workflows are presented in every experimental subsection. We will make collected instruction-following corpus, our model checkpoint, and source code publicly available once the paper is accepted.

### A.4  PRECISE DATA SCHEMA

Here we first describe the precise data format of dual-granularity alignment dataset, which consists of aligned protein sequences and property descriptions. The protein sequence represents a linear arrangement of amino acids, and the property description utilize natural language to describe comprises diverse protein attributes. Specifically, the annotated protein data is sourced from SwissProt

and TrEMBL (Bairoch & Apweiler, 2000), containing proteins with textual descriptions. The Swiss-Prot database maintains a curated collection of protein annotations that undergo rigorous manual review, while ensuring minimal redundancy among entries. The TrEMBL database contains computationally annotated protein sequences that are automatically generated through the translation of coding sequences from the EMBL (European Molecular Biology Laboratory) nucleotide corpus. A comprehensive set of protein functional annotations across diverse property domains is extracted and systematized.

- *Protein Name*: The standardized full name assigned by the UniProt consortium.
- *Function*: Diverse functional characteristics of the mature protein.
- *Subcellular Location*: Cellular localization and topological information of the protein.
- *Similarity*: Protein family classification and homology information.

To construct the biotext input for representation extraction, we concatenate available property descriptions using space delimiters, omitting any missing properties. Each property description is preceded by an annotation prefix. The detailed data schema is presented in Table 9, showing protein entry names, amino acid sequences, and corresponding property descriptions.

Subsequently, an instruction tuning stage is incorporated to endow Caduceus-Instruct with the capability of biological knowledge-based protein interpretation. To achieve this, we utilize a protein instruction dataset comprising both open-ended generation tasks and closed-set answer tasks. The open-ended generation subset is primarily built using Swiss-Prot (Bairoch & Apweiler, 2000). We include nearly all protein properties and functions available in the database. Furthermore, abundant instruction templates are manually customized. And GLM-4 (GLM et al., 2024) is employed to assist in expanding Q&A instances based on the structured annotations. The closed-set answer subset is primarily constructed from the RCSB-PDB database (Berman et al., 2000). Following the data organization established in prior work (Wu et al., 2025), we select a subset of Q&A pairs that are closely related to protein biological properties, while filtering out those pertaining to metadata (such as discovery time and experimental methods). Furthermore, we also include some protein parsing samples related to Enzyme Commission (EC) (Bairoch, 2000) and Gene Ontology (GO) (Ashburner et al., 2000) term predictions. The rigorous data format is shown in Table 10. During the instruction tuning phase, we partition the dataset into training and testing sets in the proportion of 5,231,288 / 75,977 to ensure that no data leakage occurred, following the protocol of SEPIT (Wu et al., 2025).

### A.5 THOROUGH EXPERIMENTAL DESCRIPTION

Deciphering the steerable interaction between biological and natural language lies at the heart of modern pharmacological innovation. This paradigm shift fundamentally transforms research by granting unprecedented access to nuanced language descriptions of protein properties. In Section 4.2, we incorporate 11 benchmark tasks across three task types, including localization classification, mutation effect prediction, and biological function annotation, to assess the functional engineering performance. Concretely, the localization classification task aims to predict the subcellular localization of proteins. We incorporate two such problems from DeepLoc (Almagro Armenteros et al., 2017) (*Abbr.*, Bin and Sub). The mutation effect prediction task focuses on estimating the impact of residue-level mutations on protein fitness. We utilize diverse fitness landscapes for regression evaluation, including the $\beta$-lactamase (*Abbr.*, $\beta$-lac) landscape from PEER (Xu et al., 2022), the AAV and Thermostability (*Abbr.*,Thermo) landscapes from FLIP (Dallago et al., 2021), and the Fluorescence (*Abbr.*, Flu) and Stability (*Abbr.*, Sta) landscapes from TAPE (Rao et al., 2019). Meanwhile, the biological function annotation task involves assigning multiple functional labels to elucidate proteins. The Enzyme Commission (EC) number prediction and Gene Ontology (GO) term prediction from DeepFRI (Gligorijević et al., 2021) are executed. The GO term prediction is further divided into three predictive sub-tasks: biological process (*Abbr.*, GO-BP), molecular function (*Abbr.*, GO-MF), and cellular component (*Abbr.*, GO-CC).

The growing demand for comprehensive protein analysis in fields such as pathology and drug discovery accentuates the importance of harnessing LLMs to drive advancements in general-purpose textual protein understanding (Yin et al., 2025). In Section 4.4, we include the textual protein interpretation experiment. Caduceus-Instruct employs the multi-modal aligned encoder to extract function-oriented protein representations. Furthermore, an incorporated LLM decoder generates

textual responses to interpret the molecular knowledge. To bolster experimental reliability and generalizability, we include both the open-ended and closed-set Q&A protocol for experimental assessment. Such text-based protein understanding experiment is built upon a sorted instruction dataset from Wu et al. (2025). For evaluation metrics, we employ BLEU, ROUGE, METEOR, and BERTScore for the open-ended generation task. The BERTScore is computed using Biomed-BERT (Gu et al., 2021). Accuracy is utilized to assess the closed-set answer performance. We also conduct performance comparison between Caduceus-Instruct and current powerful LLMs, incorporating DeepSeek-V3 (Liu et al., 2024a), GPT-5 (OpenAI, 2025), and Galactica-30B (Taylor et al., 2022). Such remarkable cross-modal interaction capabilities highlight Caduceus's great potential to open up new frontiers for unraveling the complexity of life with unparalleled scale and depth.

## A.6 EXTENDED ABLATION STUDIES

We compare the experimental results of MoLE utilizing varying numbers of sparsely activated LoRA experts. Specifically, we include a single LoRA expert setting, along with MoE architectures using 2/4, 2/6, and 2/8 activated LoRA experts, to facilitate comprehensive comparison. Inspired by the experimental results indicated in Table 8, our final model architecture is designed to activate the top-2 LoRA experts selected from a pool of six to optimize the trade-off between model performance and computational cost.

Table 8: Ablation study of the expert numbers within MoLE architecture.

| Settings | Sub | Thermo | EC | |
|---|---|---|---|---|
| | Acc% | Spearman's $\rho$ | AUPR | $F_{max}$ |
| single LoRA | 78.32 | 0.665 | 0.687 | 0.685 |
| MoLE-2/4 | 80.92 | 0.660 | 0.817 | 0.711 |
| **MoLE-2/6** | **83.65** | **0.682** | 0.827 | **0.808** |
| MoLE-2/8 | 83.52 | 0.669 | **0.835** | 0.801 |

Table 9: Precise data schema for the dual-granularity integration stage.

| Entry Name | Protein Sequence | Property Description |
|---|---|---|
| A0A010SA B3_9PEZI | MANSPHGGVLKDLFARDAPRQSELFAEAD KLPSLLLTERHLCDLELILNGGFSPLEGFMT EKDYNGVVKDNRLADGNLFSMPITLDVSQ QQIDTLSIKPGARITLRDLRDDRNLAILTVE DVYKPDRVKEAIEVFGSDDDTHPGVKHLFN NTNDFYVGGKLEAIQRLAHYDFLDLRFTPA ELRQHFEKLGWNKVVAFQTRNPMHRAHRE LTVRAARSQQANVLIHPVVGMTKPGDIDHF TRVRVYKALLPRYPNGMAALALLPLAMRM GGPREAIWHAIIRKNHGATHFIVGRDHAGP GKNKNGKDHYGPYDAQVAVQKYSDELGIT MVEFQEMIYIPDRDEYQPANEIAPGTHTANI SGTELRNRLKTGKEIPAWFSYPEVVKVLRE QNPLPAQKGFTIFLTGLLNSGKDQQ | **PROTEIN NAME:** Sulfate adenylyltransferase. **FUNCTION:** Catalyzes the first intracellular reaction of sulfate assimilation, forming adenosine-5'-phosphosulfate (APS) from inorganic sulfate and ATP. Plays an important role in sulfate activation as a component of the biosynthesis pathway of sulfur-containing amino acids. **SUBCELLULAR LOCATION:** Cytoplasm. **SIMILARITY:** Belongs to the APS kinase family. |
| A0A009GH C8_ACIBA | MDIFPISLKLQQQRCLIVGGGHIALRKATLL AKAGAIIDVVAPAIEDQLLQLITTTGGVSFIE AFTEKFLSTPYRLVIAATNDAEVNKTVFEQ CEARNLLVNSVDDIPHCRFMVPAIIDRSPLIV SVASNGTSPVLSRQIRTQLETSIPHGMGKLA EFSGKWRNQVKEKISNPDERRIFWENLYAS PLKEQVFNDNLDVADSMLEQALQEWKAPK GEVYLVGAGPGDPELITLKALRLMQQADV VIYDRLVSAPILELCRRDATKIYVGKARSNH SVPQEGINALLVDYAKKGKRVCRLKGGDPF IFGRGGEEIQELFQAGVPFQVVPGITAASGC SAYAGIPLTHRDYAQSVRFLTGHLKEGSPEL PWNELVYENQTLVLYMGLVGLERICEQLIA HGQRPDMPVALISKGTTPEQKVVVG | **PROTEIN NAME:** Siroheme synthase. **FUNCTION:** Multifunctional enzyme that catalyzes the SAM-dependent methylations of uroporphyrinogen III at position C-2 and C-7 to form precorrin-2 via precorrin-1. Then it catalyzes the NAD-dependent ring dehydrogenation of precorrin-2 to yield sirohydrochlorin. Finally, it catalyzes the ferrochelation of sirohydrochlorin to yield siroheme. **SIMILARITY:** Belongs to the precorrin methyltransferase family. |
| A0A024R3 24_HUMAN | MAAIRKKLVIVGDGACGKTCLLIVFSKDQF PEVYVPTVFENYVADIEVDGKQVELALWD TAGQEDYDRLRPLSYPDTDVILMCFSIDSPD SLENIPEKWTPEVKHFCPNVPIILVGNKKDL RNDEHTRRELAKMKQEPVKPEEGRDMANR IGAFGYMECSAKTKDGVREVFEMATRAAL QARRGKKKSGCLVL | **PROTEIN NAME:** Epididymis secretory sperm binding protein. |
| A0A015JW 94_RHIIW | MANIPHGGVLKDLHARDAPKKEQLLAEVE KLPSIVLSDRQLCDLELIMNGGFSPLEGFMN QEDYQSVVNNLRLKNGLLFSMPITLDVSDQ DIETLGLESKKRIVLRDPRDDAPLSILTIQDI YKPNKIEEATKVFGDDDILHPGVKYLHTQA KEFYVGGTVEAIQSPIHYDYIAHRHTPAELR AHFNKLHWTRVVAFQTRNPMHRAHRELTV RAARNRQANVLIHPVVGLTKPGDIDHYTRV RVYQALMPKYPNGMAALSLLPLAMRMGG PREAVWHAIIRKNFGNTHFIIGRDHAGPGKN SKGVDFYGPYDAQKLVAKYQDELEIEVVPF QMVTYIPDSDEYLPVDEVPEGTTTLNISGTE LRRRLRNGGHIPEWFSYPEVVKVLRETHPP RSKQGFTLFLTGYYNSGKKAIGKALEVVLN QQGGRSVTLLLGETVRHGISSELGFSRRDRD QNIARIAFVSAELTKAGAAVIAAPIA | **PROTEIN NAME:** Sulfate adenylyltransferase. **FUNCTION:** Catalyzes the first intracellular reaction of sulfate assimilation, forming adenosine-5'-phosphosulfate (APS) from inorganic sulfate and ATP. Plays an important role in sulfate activation as a component of the biosynthesis pathway of sulfur-containing amino acids. **SUBCELLULAR LOCATION:** Cytoplasm. **SIMILARITY:** Belongs to the APS kinase family. |

Table 10: Rigorous data schema for the protein instruction tuning stage.

| ProteinID | User Instruction | Answers |
|---|---|---|
| 5lau | Give me a comprehensive description of this protein. | Macrodomains are ubiquitous conserved domains that bind or transform ADP-ribose (ADPr) metabolites. In humans, they are involved in transcription, X-chromosome inactivation, neurodegeneration and modulating PARP1 signalling, making them potential targets for therapeutic agents. Unfortunately, some aspects related to the substrate binding and catalysis of MacroD-like macrodomains still remain unclear, since mutation of the proposed catalytic aspartate does not completely abolish enzyme activity. |
| 1akw | Give me the abstract of this protein. | Mutants of the electron-transfer protein flavodoxin from Desulfovibrio vulgaris were made by site-directed mutagenesis to investigate the role of glycine-61 in stabilizing the semiquinone of FMN by the protein and in controlling the flavin redox potentials. The spectroscopic properties, oxidation-reduction potentials, and flavin-binding properties of the mutant proteins, G61ANV and L, were compared with those of wild-type flavodoxin. The affinities of all of the mutant apoproteins for FMN and riboflavin were less than that of the wild-type apoprotein, and the redox potentials of the two 1-electron steps in the reduction of the complex with FMN were also affected by the mutations. Values for the dissociation constants of the complexes of the apoprotein with the semiquinone and hydroquinone forms of FMN were calculated from the redox potentials and the dissociation constant of the oxidized complex and used to derive the free energies of binding of the FMN in its three oxidation states. |
| A3QGN9 | Which protein family does this protein belong to? | This protein belongs to the transketolase family. DXPS subfamily. |
| Q09693 | What are the molecular functions of this protein? | 5'-deoxyribose-5-phosphate lyase activity; DNA-directed 5'-3' RNA polymerase activity; DNA-directed DNA polymerase activity; metal ion binding; single-strand break-containing DNA binding. |
| Q3YUY7 | What is the primary function of this protein? | Catalyzes the decarboxylation of four acetate groups of uroporphyrinogen-III to yield coproporphyrinogen-III. |
| 4bnq | Give me a comprehensive description of this protein. | Molecular replacement is the method of choice for X-ray crystallographic structure determination provided that suitable structural homologues are available in the PDB. Presently, there are 80,000 structures in the PDB (8074 were deposited in the year 2012 alone), of which 70% have been solved by molecular replacement. For successful molecular replacement the model must cover at least 50% of the total structure and the Cu03b1 r.m.s.d. between the core model and the structure to be solved must be less than 2 u00c5 |