# OpenReview forum: "Caduceus: MoE-enhanced Foundation Models Unifying Biological and Natural Language"
_ICLR.cc/2026/Conference — Submitted to ICLR 2026_

### Official Review · Reviewer_UYrt · 2025-10-31

**Soundness:** 2
**Presentation:** 3
**Contribution:** 2
**Rating:** 4
**Confidence:** 4

**Summary:**

This work propose that current multimodal pre-training combining protein sequences and diverse textual attribute descriptions suffers from severe data interference—inconsistencies in knowledge mechanisms across different attribute domains lead to degraded model performance. To address this, the authors propose Caduceus based on MoE enhancements.  Experiments are conducted to validate the proposed method.

**Strengths:**

1. The motivation of this work is clear.
2. The problem identified is important: it clearly points out the data interference caused by textual descriptions of multi-attribute proteins, which has been overlooked in previous research on multimodal proteins.
2. This work is well-written and easy to follow.

**Weaknesses:**

1. The proposed method is very simple and lacks of novelty, applying MoE-Lora to Text-protein multimodal LLM.
2. The improvement in Table 1 might be due to the addition of property desc during pre-training.
3. Several of the baselines compared were pure sequence models such as GPT-5 and DeepSeek, while the baseline chosen for the QA task was relatively weak. Models like GPT inherently lack the ability to process biological sequences, so the better performance is understandable.

**Questions:**

1. The analysis of the moe part seems inadequate; I think at least an ablation analysis of the number of experts and the rank of lora should be performed.
2. Comparison with state-of-the-art methods: While comparisons have been made with many methods, a more refined comparison can be made with other recent excellent protein-text multimodal models rather than  pure sequence LLM, under the exact same settings to highlight MoLE’s unique advantages in addressing data interference.

---

> ### Author Response · Authors · 2025-11-21
> **Response to Reviewer UYrt (1/3)**
>
> Dear Reviewer UYrt:
>
> Thanks for your comments, and we have provided detailed responses as follows.
>
> Weakness 1: The proposed method is very simple and lacks of novelty, applying MoE-Lora to Text-protein multimodal LLM.
>
> Response: **We would like to clarify that the constructed MoLE module demonstrates significant technical innovations and notable departures from conventional settings. This research exhibits substantial novelty across multiple dimensions, including collected corpus, architectural design, training objectives, and hierarchical instruction tuning framework.**
>
> **a. Substantial Technical Innovation:** The conventional MoLE framework typically assigns the encoded token without leveraging any prior contextual knowledge, and the token assignment with different characteristics exhibits a degree of stochasticity. In contrast, we propose the novel property-guided routing strategy to assigns domain-specific protein tokens to different LoRA experts. Our MoLE module receives multi-modal inputs including the protein sequence and corresponding property descriptions. The gating router assigns protein tokens to distinct LoRA experts guided by pre-aggregated prototypes of property descriptions. Furthermore, as described in Section 3.2.2, we propose Biotext-guided Static segment Reconstruction (BSR) and Property-grouped Dynamic segment Alignment (PDA) training objectives to differentially model protein tokens encoded by distinct LoRA experts. The synergy of newly proposed property-guided gate routing strategy and MoE-enhanced training objectives enables discriminative encoding of protein tokens corresponding to different properties, thereby effectively mitigating the severe data interference issue prevalent in protein multi-modality learning.
>
> **b. Broad Research Merit:** In this paper, our primary research objective is to construct foundation models that unifies biological and natural language to unlock the textual interpretation of proteins, rather than exploring the practical deployment of the MoLE module. Our study exhibits significant novelty across multiple aspects, including the collected corpus, architectural design, fine-grained training objectives, and hierarchical instruction tuning framework: (1) We collected a large-scale, multi-modal aligned pre-training corpus comprising over 250 million samples. (2) We improved the MoLE architecture with newly proposed property-guided routing strategy, coupled with two customized training objectives, to mitigate the inherent data interference issue. (3) We developed a hierarchical training framework based on multi-modal alignment and instruction tuning, empowering large language models to comprehend biological knowledge and conduct instruction-following Q&A. (4) Caduceus remarkably achieves new state-of-the-art performance on 27 out of 29 downstream evaluation metrics.
>
>
> Weakness 2: The improvement in Table 1 might be due to the addition of property desc during pre-training.
>
> Response: We would like to clarify that leveraging multi-modal alignment pre-training to enhance the performance of protein property prediction tasks has been an established and definitive research paradigm, supported by a series of well-developed research works [1-5]. In this work, during pre-training, sequences and corresponding property desc belonging to the same protein families as those in the function prediction benchmarks (Table 1) are thoroughly excluded to prevent data leakage. And We only utilize the protein encoder for downstream property prediction tasks, while the biotext encoder is excluded to ensure fair comparison. The performance improvements in Table 1 validate that multi-modal alignment effectively infuses functional information into the protein encoder, further underscoring the significant research merit of protein multi-modality learning.
>
> Question 1: The analysis of the moe part seems inadequate; I think at least an ablation analysis of the number of experts and the rank of lora should be performed.
>
> Response: **Following your advice, we have extended detailed ablation analyses on the MoLE module.**
>
> **a. Number of LoRA Experts:** We compare the experimental results of MoLE utilizing varying numbers of LoRA experts. Specifically, we include a single LoRA expert setting, along with MoE architectures using 2/4, 2/6, and 2/8 activated LoRA experts, for comprehensive comparison. Inspired by the experimental results, our final model architecture is designed to activate the top-2 LoRA experts selected from a pool of six to optimize the trade-off between model performance and computational cost.
>
>  $\qquad \qquad \qquad$Sub$\qquad$ Thermo$\quad \quad \quad \quad$EC
> | Settings | Acc% | Spearman’s ρ | AUPR | Fmax |
> |------|------:|:------:|:------:|:------:|
> | single LoRA | 78.32 | 0.665 | 0.687 | 0.685 |
> | MoLE-2/4 | 80.92 | 0.660 | 0.817 | 0.711 |
> | **MoLE-2/6** | **83.65** | **0.682** | 0.827 | **0.808** |
> | MoLE-2/8 | 83.52 | 0.669 | **0.835** | 0.801 |

---

> ### Author Response · Authors · 2025-11-21
> **Response to Reviewer UYrt (2/3)**
>
> **b. LoRA Loading ratio Rank:** We have present the loading weight distribution across distinct LoRA experts during model inference in Figure 5 (Page 10, Line 520). We investigate the activation rates across different layers, and rank the average activation ratios of the LoRA experts. The balanced activation of all experts, as evidenced by the experimental results, confirms the efficacy of the incorporated load-balancing loss (described in Section 3.1, Page 4, Line 200). This further establishes a solid foundation for the efficient parallel inference of our MoE-enhanced models.
>
> | LoRA Index | Expert 1 | Expert 2 | Expert 3 | Expert 4 | Expert 5 | Expert 6 |
> | :---------: | :------: | :------: | :------: | :------: | :------: | :------: |
> | Activate Ratio | 15.9% | 11.1% | 16.4% | 16.2% | 20.4% | 20.0% |
>
> Question 2: Comparison with state-of-the-art methods: While comparisons have been made with many methods, a more refined comparison can be made with other recent excellent protein-text multimodal models rather than pure sequence LLM, under the exact same settings to highlight MoLE’s unique advantages in addressing data interference.
> Weakness 3: Several of the baselines compared were pure sequence models such as GPT-5 and DeepSeek, while the baseline chosen for the QA task was relatively weak. Models like GPT inherently lack the ability to process biological sequences, so the better performance is understandable.
>
> Response:
>
> **a. Extensive Existing Protein Multi-Modal Baselines:** Within the protein property prediction experiment (Table 1, Page 6, Line 300) and cross-modal retrieval experiment (Table 2 & Table 3, Page 7, Line 342), we have tried our best to incorporate adequate recent excellent protein-text multi-modal models to facilitate comprehensive comparison, including ProtST [1], ProteinCLAP [3], OntoProtein [4], ProtT3 [6], and SEPIT [7]. Caduceus remarkably achieves new state-of-the-art performance on 27 out of 29 downstream evaluation metrics, outperforming all these powerful protein multi-modality models.
>
> **b. Ablative Baseline to Highlight MoLE's Unique Advantage:** For the biological knowledge-based Q&A experiment, we would like to clarify that only very limited existing models have undergone protein multi-modal instruction tuning to enable the highly challenging task of text-based protein knowledge Q&A. And we have incorporated multiple protein instruction tuning models, including ProtT3 [6], SEPIT [7], InstructProtein [8] and Galactica [9], as baseline models in Table 4. Additionally, We have added an ablative baseline that shares the identical setting with Caduceus, with the sole exclusion of the MoLE module, for comparison. As anticipated, the baseline model demonstrate suboptimal performance. This further reaffirms the inherent data interference issue, which is effectively mitigated by the improved MoLE module.
>
> Table 4. Evaluation performance on the text-based protein understanding experiment.
>
> | Models               | BLEU-2 | BLEU-4 | ROUGE-1 | ROUGE-2 | ROUGE-L | METEOR | BERT-P | BERT-R | BERT-F1 | Accuracy |
> | :----------- | :----: | :----: | :-----: | :-----: | :-----: | :----: | :----: | :----: | :-----: | :------: |
> | Galactica            |  0.43  |  0.01  |  3.49   |  0.41   |  2.67   |  2.44  | 85.79  | 82.61  |  84.08  |  39.15%  |
> | BioT5+               |  3.88  |  1.92  |  12.12  |  4.88   |  10.37  | 14.26  | 85.14  | 85.93  |  85.48  |    —     |
> | InstructProtein      |  5.50  |  2.97  |  14.80  |  5.68   |  13.76  | 13.17  | 85.34  | 85.92  |  85.57  |  48.37%  |
> | OpenLLaMA            | 36.19  | 30.65  |  48.33  |  36.52  |  45.53  | 49.01  | 92.92  | 91.87  |  92.35  |  71.77%  |
> | LLaMA2-Chat          | 57.02  | 49.47  |  70.80  |  57.24  |  67.78  | 65.96  | 94.95  | 95.17  |  95.05  |  71.68%  |
> | ProtT3               | 58.58  | 50.27  |  70.28  |  56.27  |  67.93  | 66.38  | 94.92  | 95.28  |  95.51  |  77.98%  |
> | SEPIT                | 58.43  | 51.04  |  72.34  |  58.77  |  69.13  | 67.91  | 95.32  | 95.59  |  95.44  |  79.05%  |
> | Caduceus-w/o-MoLE    | 56.49  | 50.28  |  71.75  |  58.00  |  68.92  | 65.15  | 95.19  | 95.25  |  95.37  |  72.04%  |
> | Caduceus-650M-Instruct | $\underline{62.74}$  | $\underline{56.39}$  |  $\underline{76.86}$  |  $\underline{63.01}$  |  $\underline{74.45}$  | $\underline{70.13}$  | **95.92**  | $\underline{96.00}$  |  $\underline{95.87}$  |  $\underline{80.66}$%  |
> | Caduceus-3B-Instruct | **68.48**  | **62.09**  |  **81.23**  |  **69.05**  |  **79.94**  | **73.75**  | $\underline{95.77}$  | **96.59**  |  **96.48**  |  **83.84%**  |

---

> ### Author Response · Authors · 2025-11-21
> **Response to Reviewer UYrt (3/3)**
>
> **c. Newly incorporated Protein Multi-Modal Experiments:** Furthermore, we have added two novel experiments (amino acid contact prediction & protein-protein interaction identification) in the newly incorporated Section 4.5 (Page 9, Line 460) to facilitate more holistic comparison with recent excellent protein multi-modal models. Notably, OntoProtein [4], KeAP [10], and Kara [11] are powerful baseline models extensively pre-trained on protein and text multi-modal data. As presented in Table 5 and Table 6, we are delighted to report that our model achieves new state-of-the-art performance on all 18 evaluation metrics.
>
> Table 5: Experimental results of amino acid contact prediction.
>
> $\qquad \qquad \qquad \qquad \quad$ 6 $\leq$ seq $\leq$ 12 $\qquad$ 12 $\leq$ seq $\leq$ 24 $\qquad \quad$ 24 $\leq$ seq
> | Models | P@L | P@L/2 | P@L/5 | P@L | P@L/2 | P@L/5 | P@L | P@L/2 | P@L/5 |
> |---|---:|:---:|:---|---:|:---:|:---|---:|:---:|:---|
> | LSTM | 0.26 | 0.36 | 0.49 | 0.20 | 0.26 | 0.34 | 0.20 | 0.23 | 0.27 |
> | ResNet | 0.25 | 0.34 | 0.46 | 0.28 | 0.25 | 0.35 | 0.10 | 0.13 | 0.17 |
> | Transformer | 0.28 | 0.35 | 0.46 | 0.19 | 0.25 | 0.33 | 0.17 | 0.20 | 0.24 |
> | ProtBert | 0.30 | 0.40 | 0.52 | 0.27 | 0.35 | 0.47 | 0.20 | 0.26 | 0.34 |
> | ESM-1b | 0.38 | 0.48 | 0.62 | 0.33 | 0.43 | 0.56 | 0.26 | 0.34 | 0.45 |
> | ESM-2 | 0.40 | 0.50 | 0.62 | 0.35 | 0.44 | 0.56 | 0.27 | 0.35 | 0.45 |
> | OntoProtein | 0.37 | 0.46 | 0.57 | 0.32 | 0.40 | 0.50 | 0.24 | 0.31 | 0.39 |
> | KeAP | 0.41 | 0.51 | 0.63 | 0.36 | 0.45 | 0.54 | 0.28 | 0.35 | 0.43 |
> | Kara | 0.45 | 0.55 | 0.65 | 0.39 | 0.48 | 0.59 | $\underline{0.31}$ | $\underline{0.39}$ | 0.48 |
> | Caduceus-650M | $\underline{0.47}$ | $\underline{0.58}$ | $\underline{0.67}$ | $\underline{0.43}$ | $\underline{0.51}$ | $\underline{0.61}$ | 0.30 | 0.36 | $\underline{0.50}$ |
> | Caduceus-3B | **0.50** | **0.62** | **0.69** | **0.44** | **0.55** | **0.63** | **0.35** | **0.47** | **0.53** |
>
> Table 6: Evaluation performance of protein-protein interaction identification.
>
> $\qquad \qquad \qquad \qquad \qquad$ SHS27K $\qquad \qquad$ SHS148K $\qquad \qquad$ STRING
> | Models | BFS | DFS | Avg | BFS | DFS | Avg | BFS | DFS | Avg |
> |---|---:|:---:|:---|---:|:---:|:---|---:|:---:|:---|
> | DNN-PPI | 48.09 | 54.34 | 51.22 | 57.40 | 58.42 | 57.91 | 53.05 | 64.94 | 59.00 |
> | DPPI | 41.43 | 46.12 | 43.77 | 52.12 | 52.03 | 52.08 | 56.68 | 66.82 | 61.75 |
> | PIPR | 44.48 | 57.80 | 51.14 | 61.83 | 63.98 | 62.91 | 55.65 | 67.45 | 61.55 |
> | GNN-PPI | 63.81 | 74.72 | 69.27 | 71.37 | 82.67 | 77.02 | 78.37 | 91.07 | 84.72 |
> |ProtBert | 70.94 | 73.36 | 72.15 | 70.32 | 78.86 | 74.59 | 67.61 | 87.44 | 77.53 |
> | ESM-1b | 74.92 | 78.83 | 76.88 | 77.49 | 82.13 | 79.31 | 78.54 | 88.59 | 83.57 |
> | ESM-2 | 75.05 | 79.55 | 77.30 | 77.19 | 83.34 | 80.26 | 81.32 | 89.19 | 85.30 |
> | OntoProtein | 72.26 | 78.89 | 75.58 | 75.23 | 77.52 | 76.38 | 76.71 | 91.45 | 84.08 |
> | KeAP | 78.58 | 77.54 | 78.06 | 77.22 | 84.74 | 80.98 | 81.44 | 89.77 | 85.61 |
> | Kara | 81.18 | 78.85 | 80.01 | 79.62 | 86.02 | 82.82 | $\underline{82.73}$ | 92.46 | $\underline{87.59}$ |
> | Caduceus-650M | $\underline{82.13}$ |  $\underline{79.26}$ |  $\underline{80.69}$ |  $\underline{82.25}$ |  $\underline{87.63}$ |  $\underline{84.94}$ | 81.25 | **92.84** | 87.04 |
> | Caduceus-3B | **84.51** | **80.25** | **82.38** | **83.64** | **89.52** | **86.58** | **83.27** | $\underline{92.63}$ | **87.95** |
>
>
> References:
>
> [1] ProtST: Multi-modality learning of protein sequences and biomedical texts. In ICML 2023.
>
> [2] ProtCLIP: Function-informed protein multi-modal learning. In AAAI 2025.
>
> [3] A text-guided protein design framework. Nature Machine Intelligence, pages 1–12, 2025.
>
> [4] OntoProtein: Protein Pretraining With Gene Ontology Embedding. In ICLR 2022.
>
> [5] Retrieval-Augmented Language Model for Knowledge-aware Protein Encoding. In ICML, 2025.
>
> [6] ProtT3: Protein-to-text generation for text-based protein understanding. In ACL, 2024.
>
> [7] Towards General-Purpose Protein Understanding with LLMs. In ACM SIGKDD, 2025.
>
> [8] InstructProtein: Aligning human and protein language via knowledge instruction. In ACL, 2024.
>
> [9] Galactica: A large language model for science. arXiv, 2022.
>
> [10] Protein Representation Learning via Knowledge Enhanced Primary Structure Reasoning. In ICLR, 2023.
>
> [11] Retrieval-Augmented Language Model for Knowledge-aware Protein Encoding. In ICML, 2025.

---

### Official Review · Reviewer_x9cp · 2025-11-01

**Soundness:** 2
**Presentation:** 3
**Contribution:** 2
**Rating:** 4
**Confidence:** 4

**Summary:**

The paper introduces Caduceus, a multimodal protein model that links biological sequences with natural language. It leverages on the Mixture of LoRA Experts framework, using a property-guided router to send protein tokens to specialized experts and reduce interference between biological domains. Training has two stages: dual-granularity alignment, which connects protein and text representations at both global and local levels, and instruction tuning, which enables question answering with a language model decoder. Caduceus achieves strong results on multiple protein and text benchmarks, showing that domain-specific routing and multimodal training improve performance and interpretability.

**Strengths:**

- Clear motivation: Identifies and quantifies the “data interference” problem across protein property domains.

- Effective adaptation: Extends the Mixture of LoRA Experts [1] with a biologically informed gating router that routes protein tokens by property.

- Dual-granularity learning: Combines global and local protein-text alignment to capture both sequence-level and residue-level meaning.

- Hierarchical pipeline: Two-stage training (alignment, instruction tuning) connects representation learning with natural language generative modeling.

- Good empirical results: Achieves state-of-the-art performance across 15 benchmarks with solid ablations.

- Clarity and presentation: Well-structured paper with intuitive figures and transparent methodology.

[1] Wu et al., Mixture of LoRA Experts, ICLR 2024, https://arxiv.org/pdf/2404.13628

**Weaknesses:**

- Limited novelty: Core MoLE mechanism is based on prior work (e.g., [1], as already cited in this paper), so the main technical innovation of this paper seems to be the domain adaptation and the pipeline. I would suggest authors clarify and distinguish the main technical innovation of this paper given the prior work.

- Concern about data overlap: Pretraining and evaluation datasets may share entries, risking leakage. I would suggest the authors discuss how they ensure there is not data leakage

- Claim issue: In the conclusion, line 478, it is writen that "we propose the Mixture of LoRA Experts (MoLE) to effectively...". This may confuse some readers into thinking its a claim of introducing MoLE itself, while MoLE has already been introduced and studied before. The authors may want to rewrite this to avoid potential confusions/concerns.

- [Minor] Concern related to fair comparison: Instruction-tuned model is compared to zero-shot general LLMs. While this is a reasonable baseline, this may not be a fair comparison.

- [Minor] Gap in validation: It would be interesting to see causal or mechanistic tests (e.g., in silico mutagenesis, binding effect studies).

[1] Wu et al., Mixture of LoRA Experts, ICLR 2024, https://arxiv.org/pdf/2404.13628

**Questions:**

1. Can the authors clarify and distinguish their main technical innovation given the prior work?

2. Can the authors provide a discussion on how they ensure there is not data leakage?

---

> ### Author Response · Authors · 2025-11-18
> **Response to Reviewer x9cp (1/2)**
>
> Dear Reviewer x9cp:
>
> Thanks for the your feedback. We have provided detailed responses to the comments.
>
> Weakness 1: Core MoLE mechanism is based on prior work (e.g., [1], as already cited in this paper), so the main technical innovation of this paper seems to be the domain adaptation and the pipeline. I would suggest authors clarify and distinguish the main technical innovation of this paper given the prior work.
> Question 1: Can the authors clarify and distinguish their main technical innovation given the prior work?
>
> Response:
>
> **a. Novel Gate Routing Strategy**: In terms of the technical details of the MoLE module, the prior MoLE framework [1] directly assigns encoded uni-modal tokens to various experts, and the token assignment with different characteristics exhibits a degree of stochasticity. In contrast, we propose the property-guided gating router to assigns domain-specific protein tokens to different LoRA experts. Our MoLE module receives multi-modal inputs and assigns the protein tokens to distinct experts guided by pre-aggregated prototypes of property descriptions. Furthermore, as described in Section 3.2.2, we propose Biotext-guided Static segment Reconstruction (BSR) and Property-grouped Dynamic segment Alignment (PDA) training objectives to differentially model protein tokens encoded by distinct experts. The synergy of newly proposed gate routing strategy and training objectives enables discriminative encoding of protein tokens corresponding to different properties, thereby effectively mitigating the severe data interference issue prevalent in protein multi-modality learning.
>
> **b. Holistic Research Contributions**: In terms of the overall research approach, the prior work [1] simply focuses on investigating the optimal LoRA composition architecture. In contrast, our work aims to construct foundation models that unifies biological and natural language to unlock the textual interpretation of proteins, with comprehensive innovations across collected corpus, architectural design, and training paradigm. Our main technical contributions are summarized as follows: (1) We collected a large-scale, multi-modal aligned pre-training corpus comprising over 250 million samples. (2) We proposed a novel property-guided gate routing scheme, coupled with two customized training objectives, to mitigate the inherent data interference issue. (3) We developed a hierarchical training framework based on multi-modal alignment and instruction tuning, empowering large language models to comprehend biological knowledge and conduct instruction-following question-answering. (4) Caduceus remarkably achieves new state-of-the-art performance on 27 out of 29 downstream evaluation metrics.
>
> Weakness 2: Concern about data overlap: Pretraining and evaluation datasets may share entries, risking leakage. I would suggest the authors discuss how they ensure there is not data leakage.
> Question 2: Can the authors provide a discussion on how they ensure there is not data leakage?
>
> Response:
> Within the hierarchical training framework, we adhere to the standard data processing pipeline established in prior classical research works [2, 3] to ensure the absence of data leakage. Relevant data details are provided in the Supplementary Materials.
>
> a. In the multi-modality integration stage, pre-training sequences belonging to the same protein families as those in the function prediction benchmarks (*i.e.*, with maximum amino acid similarity < 40% using the CD-HIT clustering algorithm) were excluded to avoid data leakage following ProtST [2].
>
> b. During the instruction tuning phase, we employed a proportional split (5,231,288 / 75,977) to partition the data into training and test sets, following the protocol of SEPIT [3] to ensure that no data leakage occurred.
>
> Weakness 3: Claim issue: In the conclusion, line 478, it is writen that "we propose the Mixture of LoRA Experts (MoLE) to effectively...". This may confuse some readers into thinking its a claim of introducing MoLE itself, while MoLE has already been introduced and studied before. The authors may want to rewrite this to avoid potential confusions/concerns.
>
> Response: The corresponding expressions have been reformulated to enhance the reader's accurate comprehension following your kind advice (Page 10, Line 532).
>
>
> References:
>
> [1] Mixture of LoRA Experts. ICLR 2024.
>
> [2] ProtST: Multi-modality learning of protein sequences and biomedical texts. ICML 2023.
>
> [3] Towards General-Purpose Protein Understanding with LLMs. SIGKDD 2025.

---

> > ### Author Response · Authors · 2025-11-18
> > **Response to Reviewer x9cp (2/2)**
> >
> > Weakness 4: [Minor] Concern related to fair comparison: Instruction-tuned model is compared to zero-shot general LLMs. While this is a reasonable baseline, this may not be a fair comparison.
> >
> > Response:
> >
> > a. Given OpenAI's claim that the GPT series of LLMs possess some understanding of biological knowledge, we consider the introduction of powerful LLMs as interesting baselines. The suboptimal performance of the GPT series further underscores the unique research value of our work in infusing biological knowledge into LLMs through protein instruction tuning.
> >
> > b. In the manuscript, we have specified the experimental settings of the zero-shot and instruction-tuning models to ensure readers' accurate interpretation.
> >
> > Weakness 5: [Minor] Gap in validation: It would be interesting to see causal or mechanistic tests (e.g., in silico mutagenesis, binding effect studies).
> >
> > Response: Thanks for your valuable suggestion, and we agree that **incorporating a mechanistic test** could enhance the comprehensiveness of experimental validation. Specifically, following your advice, Caduceus is queried to identify proteins with potentially high binding affinity to heme. Then, an exhaustive all-versus-all search is performed within the Gene Ontology [4] database based on the representation similarities. The top-3 ranked proteins are 1YHU, 2N91, and 5VPR. We then utilize AutoDock [5] to accurately predict the binding affinity values (*the lower the better*). The binding affinity of proteins retrieved by our model is significantly stronger than that of randomly selected proteins, which further validates Caduceus' capability to comprehend both biological and natural language.
> >
> > | candidates |         top-3 retrieved proteins     |    30 randomly selected proteins     |
> > |    :----   |    :----:   |    :----:   |
> > | binding affinity ↓ |        **-7.53 kcal/mol** |                 -5.16 kcal/mol  |
> >
> > References:
> >
> > [4] Gene Ontology: tool for the unification of biology. Nature Genetics, 25(1):25–29, 2000.
> >
> > [5] AutoDock Vina: improving the speed and accuracy of docking with a new scoring function, efficient optimization, and multithreading. Journal of Computational Chemistry, 31(2):455–461, 2010.

---

### Official Review · Reviewer_xRjq · 2025-11-01

**Soundness:** 3
**Presentation:** 3
**Contribution:** 3
**Rating:** 6
**Confidence:** 4

**Summary:**

This paper notices that existing protein language models struggle to deal with proteins' textual descriptions in heterogeneous domains. To mitigate this problem, this paper proposes a mixture-of-LoRA method to fuse knowledge of different domains. Specifically, this paper proposes a hierarchical pretraining method and property-guided gating router. The proposed multi-task instruction tuning also shows effectiveness on benchmark datasets.

**Strengths:**

1. Overall, the paper is well written, with figures as visual illustrations. The Introduction section clearly explains the motivation behind the method. It also makes a comparison to existing methods and identifies their drawbacks.

2. Using mixture-of-LoRA experts for multi-modal protein language modeling is novel to me, and experiments also show the effectiveness of the proposed method.

3. Experiments are conducted on multiple benchmark datasets. Both quantitative and qualitative tasks are conducted to comprehensively show the effectiveness of the proposed method.

**Weaknesses:**

1. Usually when we do experiments, we encourage authors to repeat the same experimental setting multiple times and report both mean and standard deviation. However, this paper shows mean but not stddev, which is difficult for readers to judge how significantly the proposed method outperforms baselines.

2. Though this paper proposes an interesting method, it misses to mention and compare to a highly related existing work [1], which uses retrieval-augmented method to integrate knowledge graph with textual descriptions into proteins' amino acid sequences for a multi-modal representation learning.

[1] Zhang, J., Zhang, D. C., Liang, S., Li, Z., Ying, R., & Shao, J. Retrieval-Augmented Language Model for Knowledge-aware Protein Encoding. In Forty-second International Conference on Machine Learning.

**Questions:**

N/A

---

> ### Author Response · Authors · 2025-11-20
> **Response to Reviewer xRjq (1/2)**
>
> Dear Reviewer xRjq:
>
> We appreciate your insightful feedback and have provided comprehensive responses to each of your comments.
>
> Weakness 1: Usually when we do experiments, we encourage authors to repeat the same experimental setting multiple times and report both mean and standard deviation. However, this paper shows mean but not stddev, which is difficult for readers to judge how significantly the proposed method outperforms baselines.
>
> Response:
>
> a. Following your advice, we have supplemented the standard deviation metrics in the cross-modal retrieval experiments presented in Section 4.3 to more comprehensively demonstrate the superiority of our model.
>
> $\qquad \qquad \qquad \quad $ Protein-to-Text $\qquad \quad $ Text-to-Protein
> |  Models  |  Accuracy  |  Recall@20  |  Accuracy  |   Recall@20  |
> |---|:---:|:---:|:---:|:---:|
> |  ProtST  |  5.5 (2.61)  |  41.6 (3.35)  |  5.8 (2.79)  |  43.3 (3.84)  |
> |  ProteinCLAP  |  39.0 (3.02)  |  89.4 (2.53)  |  39.3 (3.59)  |  89.7 (2.67)  |
> |  ProtT3  |  55.8 (1.31)  |  91.7 (0.90)  |  55.6 (1.77)  |  91.7 (0.83)  |
> |  Caduceus  |  **62.1 (0.75)**  |  **97.5 (0.41)**  |  **63.6 (1.06)**  |  **97.9 (0.49)**  |
>
> b. For the remaining experiments in Section 4.2 and Section 4.4, we directly adopt the baseline performance metrics reported in prior classical studies [1, 2]. However, the stddev values are not documented in previous works [1, 2], and the computational cost associated with re-implementing all baselines would be prohibitively inefficient. Within existing experiments, Caduceus remarkably achieves new state-of-the-art performance on 27 out of 29 downstream evaluation metrics. Furthermore, we have added two novel experiments (*see detailed description in our response to weakness 2*), and Caduceus once again attains optimal results across all 18 evaluation metrics. The consistent and substantial performance enhancements adequately validate the superior performance of our model.
>
> Weakness 2: Though this paper proposes an interesting method, it misses to mention and compare to a highly related existing work [3], which uses retrieval-augmented method to integrate knowledge graph with textual descriptions into proteins' amino acid sequences for a multi-modal representation learning.
>
> Response:
>
> In accordance with your meticulous suggestion, we have added the citation of Kara [3] in the Related Work (Page 3, Line 114). Moreover, we have incorporated representative experiments from [3] to holistically compare the performance of Caduceus and Kara. **We are excited to report that our model achieves new state-of-the-art performance on all 18 evaluation metrics, surpassing the previous best performance established by Kara.**
>
> **a. Amino Acid Contact Prediction:** Contact prediction task aims to predict whether two amino acids within a protein are in contact, which is a protein token-level classification task. We employs the same experimental setting in Section 4.1 of Kara [3]. The best performance is highlighted in bold, and the second-best result is underlined. Relevant experimental results have been compiled into Table 5 of the manuscript (Page 9, Line 474).
>
> $\qquad \qquad \qquad \qquad \quad$ 6 $\leq$ seq $\leq$ 12 $\qquad$ 12 $\leq$ seq $\leq$ 24 $\qquad \quad$ 24 $\leq$ seq
> | Models | P@L | P@L/2 | P@L/5 | P@L | P@L/2 | P@L/5 | P@L | P@L/2 | P@L/5 |
> |---|---:|:---:|:---|---:|:---:|:---|---:|:---:|:---|
> | LSTM | 0.26 | 0.36 | 0.49 | 0.20 | 0.26 | 0.34 | 0.20 | 0.23 | 0.27 |
> | ResNet | 0.25 | 0.34 | 0.46 | 0.28 | 0.25 | 0.35 | 0.10 | 0.13 | 0.17 |
> | Transformer | 0.28 | 0.35 | 0.46 | 0.19 | 0.25 | 0.33 | 0.17 | 0.20 | 0.24 |
> | ProtBert | 0.30 | 0.40 | 0.52 | 0.27 | 0.35 | 0.47 | 0.20 | 0.26 | 0.34 |
> | ESM-1b | 0.38 | 0.48 | 0.62 | 0.33 | 0.43 | 0.56 | 0.26 | 0.34 | 0.45 |
> | ESM-2 | 0.40 | 0.50 | 0.62 | 0.35 | 0.44 | 0.56 | 0.27 | 0.35 | 0.45 |
> | OntoProtein | 0.37 | 0.46 | 0.57 | 0.32 | 0.40 | 0.50 | 0.24 | 0.31 | 0.39 |
> | KeAP | 0.41 | 0.51 | 0.63 | 0.36 | 0.45 | 0.54 | 0.28 | 0.35 | 0.43 |
> | Kara | 0.45 | 0.55 | 0.65 | 0.39 | 0.48 | 0.59 | $\underline{0.31}$ | $\underline{0.39}$ | 0.48 |
> | Caduceus-650M | $\underline{0.47}$ | $\underline{0.58}$ | $\underline{0.67}$ | $\underline{0.43}$ | $\underline{0.51}$ | $\underline{0.61}$ | 0.30 | 0.36 | $\underline{0.50}$ |
> | Caduceus-3B | **0.50** | **0.62** | **0.69** | **0.44** | **0.55** | **0.63** | **0.35** | **0.47** | **0.53** |

---

> ### Author Response · Authors · 2025-11-20
> **Response to Reviewer xRjq (2/2)**
>
> **b. Protein-Protein Interaction Identification:** Protein-protein interaction (PPI) identification
> aims to predict the interaction state of protein pairs and is a sequence-level classification task. And we follow the identical experimental setting presented in Section 4.2 of Kara [3]. The corresponding experimental results have been compiled into Table 6 of the manuscript (Page 9, Line 474).
>
> $\qquad \qquad \qquad \qquad \qquad$ SHS27K $\qquad \qquad$ SHS148K $\qquad \qquad$ STRING
> | Models | BFS | DFS | Avg | BFS | DFS | Avg | BFS | DFS | Avg |
> |---|---:|:---:|:---|---:|:---:|:---|---:|:---:|:---|
> | DNN-PPI | 48.09 | 54.34 | 51.22 | 57.40 | 58.42 | 57.91 | 53.05 | 64.94 | 59.00 |
> | DPPI | 41.43 | 46.12 | 43.77 | 52.12 | 52.03 | 52.08 | 56.68 | 66.82 | 61.75 |
> | PIPR | 44.48 | 57.80 | 51.14 | 61.83 | 63.98 | 62.91 | 55.65 | 67.45 | 61.55 |
> | GNN-PPI | 63.81 | 74.72 | 69.27 | 71.37 | 82.67 | 77.02 | 78.37 | 91.07 | 84.72 |
> |ProtBert | 70.94 | 73.36 | 72.15 | 70.32 | 78.86 | 74.59 | 67.61 | 87.44 | 77.53 |
> | ESM-1b | 74.92 | 78.83 | 76.88 | 77.49 | 82.13 | 79.31 | 78.54 | 88.59 | 83.57 |
> | ESM-2 | 75.05 | 79.55 | 77.30 | 77.19 | 83.34 | 80.26 | 81.32 | 89.19 | 85.30 |
> | OntoProtein | 72.26 | 78.89 | 75.58 | 75.23 | 77.52 | 76.38 | 76.71 | 91.45 | 84.08 |
> | KeAP | 78.58 | 77.54 | 78.06 | 77.22 | 84.74 | 80.98 | 81.44 | 89.77 | 85.61 |
> | Kara | 81.18 | 78.85 | 80.01 | 79.62 | 86.02 | 82.82 | $\underline{82.73}$ | 92.46 | $\underline{87.59}$ |
> | Caduceus-650M | $\underline{82.13}$ |  $\underline{79.26}$ |  $\underline{80.69}$ |  $\underline{82.25}$ |  $\underline{87.63}$ |  $\underline{84.94}$ | 81.25 | **92.84** | 87.04 |
> | Caduceus-3B | **84.51** | **80.25** | **82.38** | **83.64** | **89.52** | **86.58** | **83.27** | $\underline{92.63}$ | **87.95** |
>
>
> **c. Novel Biological Knowledge-based Q&A Capability:** It is important to note that both Kara [3] and KeAP [4] inject function annotation information into protein encoders based on Protein Knowledge Graphs (PKGs). The absence of generative LLMs prevents them from achieving user-following Q&A. In contrast, following the multi-modal alignment process, we subsequently employ the instruction tuning technique to infuse protein-specific knowledge into LLaMA3-8B. The innovative protein instruction tuning phase endows Caduceus-Instruct with novel capabilities for interpreting proteins using natural language, which is not possessed by Kara [3] or KeAP [4].
>
>
> References:
>
> [1] ProtST: Multi-modality learning of protein sequences and biomedical texts. In International Conference on Machine Learning, 2023.
>
> [2] Towards General-Purpose Protein Understanding with LLMs. In ACM SIGKDD International Conference on Knowledge Discovery and Data Mining, 2025.
>
> [3] Retrieval-Augmented Language Model for Knowledge-aware Protein Encoding. In International Conference on Machine Learning, 2025.
>
> [4] Protein Representation Learning via Knowledge Enhanced Primary Structure Reasoning. In International Conference on Learning Representations, 2023.

---

### Official Review · Reviewer_6ui8 · 2025-11-09

**Soundness:** 3
**Presentation:** 3
**Contribution:** 3
**Rating:** 6
**Confidence:** 2

**Summary:**

This paper presents Caduceus, a multimodal foundation model that unifies large language models and protein language models. Traditional approaches to fusing natural language and protein language domains often suffer from data interference issues. The authors attempt to mitigate this by adopting a mixture of experts (MoE) architecture, where the gating mechanism tries to distinguish between language tokens and protein tokens. Details of the alignment and instruction tuning processes are provided for the development of Caduceus. Experimental results demonstrate the effectiveness of Caduceus across both natural language and life science domains.

**Strengths:**

- The paper is well written and well motivated overall.
- Unifying life science and natural language data modalities is an important research direction to pursue.
- The proposed Caduceus method is clearly presented and easy to follow.
- The performance of Caduceus appears to be promising.

**Weaknesses:**

- The development process of Caduceus seems to heavily leverage the QA data described in Section 4.1, which makes it challenging for the model to scale.
- It is not stated in the paper whether the developed QA dataset will be publicly released for future research, which I strongly encourage the authors to do.
- When scaling the model size from 650M to 3B, the accuracy gains appear to be marginal, so I am uncertain about the scaling behavior of Caduceus.

**Questions:**

- DNA sequence models have started to emerge recently, possibly as alternatives to protein language models. For instance, the recent release of AlphaGenome attempts to use DNA sequences to perform downstream tasks directly. I wonder whether Caduceus can be extended to also support DNA sequences.

---

> ### Author Response · Authors · 2025-11-18
> **Response to Reviewer 6ui8 (1/2)**
>
> Dear Reviewer 6ui8:
>
> Thanks for your valuable comments. We have provided detailed responses to your commented weaknesses and questions.
>
> Weakness 1: The development process of Caduceus seems to heavily leverage the QA data described in Section 4.1, which makes it challenging for the model to scale.
>
> Response: As described in Section 4.1, we incorporated a wide range of protein databases during the collection of Q&A data, including UniProtKB [1], RCSB-PDB [2], Enzyme Commission [3], and Gene Ontology [4] databases, to ensure strong scalability of our model. This enables Caduceus to operate without reliance on domain-specific knowledge, thereby allowing it to address a broad range of biological questions across diverse paradigms, as illustrated in Figure 4.
>
> Weakness 2: It is not stated in the paper whether the developed QA dataset will be publicly released for future research, which I strongly encourage the authors to do.
>
> Response: In accordance with fundamental principles of scientific research, we will make the model checkpoints, collected training corpus, and source code publicly available to advance the development of the AI for Science research community. We have included the open-source statement in the Abstract of the manuscript (Page 1, Line 28).
>
> Weakness 3: When scaling the model size from 650M to 3B, the accuracy gains appear to be marginal, so I am uncertain about the scaling behavior of Caduceus.
>
> Response:
>
> a. As illustrated in Table 1, Caduceus-3B achieved significantly superior performance over Caduceus-650M across three task types under the linear probing setting. We believe that the consistent performance improvement observed across diverse downstream tasks strongly demonstrates the robust generalization capability and scalability of our model.
>
> $\qquad\qquad\qquad\qquad\qquad\qquad\qquad\qquad\qquad\quad$EC$\qquad\qquad$GO_BP$\qquad\qquad$GO_MF$\qquad\quad$GO_CC
> | Models  | β-lac | AAV  | Flu  | $AUPR$  | $F_{max}$ | $AUPR$ | $F_{max}$ | $AUPR$  | $F_{max}$ | $AUPR$  | $F_{max}$ |
> |:-----------------|:-----:|:----:|:----:|:----:|:----:|:-------:|:-------:|:-------:|:-------:|:-------:|:-------:|
> | ProtBERT | 0.616 | 0.209| 0.339| 0.028| 0.089| 0.130   | 0.245 | 0.053 | 0.120 | 0.143   | 0.296   |
> | OntoProtein| 0.471 | 0.217| 0.432| 0.411| 0.417| 0.243   | 0.345   | 0.418   | 0.383 | 0.346   | 0.465   |
> | ESM-1b | 0.528 | 0.454| 0.430| 0.649| 0.642| 0.309   | 0.403  | 0.557 | 0.528 | 0.404   | 0.504   |
> | ESM-2 | 0.559 | 0.374| 0.456| 0.711| 0.694| 0.311   | 0.412   | 0.577   | 0.547 | 0.404 | 0.519   |
> | ProtST | 0.565 | 0.398| 0.499| 0.810| 0.784| 0.358   | 0.458   | 0.643   | 0.601 | 0.451 | 0.546   |
> | Caduceus-650M  | 0.565 | 0.532| 0.503| 0.827| 0.808| 0.453   | 0.556   | 0.635 | 0.598 | 0.492   | 0.586   |
> |**Caduceus-3B** | **0.637** | **0.602**| **0.585**| **0.871**| **0.859**| **0.462** | **0.567**   | **0.673**   | **0.656**   | **0.535**   | **0.608**   |
> | **Scaling Gains**|**12.7%**|**13.2%**|**16.3%**|**5.3%**|**6.3%**|**2.0%** |**2.0%** |**6.0%** |**9.7%** |**8.7%** |**3.8%** |
>
> b. We understand your concern regarding average the marginal scaling performance improvements in Table 1 under the full fine-tuning setting (although Caduceus-3B still delivers **9.64% and 15.73% AUPR improvements** on GO-BP and GO-CC over Caduceus-650M utilizing full fine-tuning paradigm). However, we contend that the marginal performance gains are principally due to the inherent limitations of the utilized biological evaluation datasets, rather than a deficiency in the model's generalization capability. As documented in the relevant literature [5-8], the inherent noise introduced by wet-lab procedures during biological data collection, together with the prevalence of relatively simplistic task paradigms, has contributed to the observed trend of performance saturation under full fine-tuning setting.
>
> c. To fully address your concerns and substantiate our assertion outlined in b, **we have newly conducted further evaluation of Caduceus-3B on the benchmark task presented in Table 4, comparing it with Caduceus-650M.** The evaluation results demonstrated the remarkable performance improvements on the more complex biological knowledge-based Q&A task.
>
> | Models |  BLEU-2  |  BLEU-4  | ROUGE-1  | ROUGE-2  | ROUGE-L  |  METEOR  |
> |:---------------|:--------:|:--------:|:--------:|:--------:|:--------:|:--------:|
> | InstructProtein |   5.50   |  2.97 |  14.80 |   5.68 |  13.76 | 13.17 |
> | OpenLLaMA |  36.19   |  30.65 |  48.33 |  36.52   |  45.53 |  49.01 |
> | LLaMA2-Chat |  57.02   |  49.47 |  70.80 |  57.24 |  67.78 |  65.96 |
> | ProtT3 |  58.58 |  50.27 |  70.28 |  56.27 |  67.93 |  66.38 |
> | SEPIT |  58.43 |  51.04 |  72.34 |  58.77 |  69.13 |  67.91 |
> | Caduceus-650M-Instruct | 62.74| 56.39| 76.86| 63.01| 74.45| 70.13|
> | **Caduceus-3B-Instruct** | **68.48**| **62.09**| **81.23**| **69.05**| **79.94**| **73.75**|
> | **Scaling Gains** | **9.15%**|**10.11%**| **5.68%**| **9.60%**| **7.38%**| **5.15%**|

---

> ### Author Response · Authors · 2025-11-27
> **Response to Reviewer 6ui8 (2/2)**
>
> Question 1: DNA sequence models have started to emerge recently, possibly as alternatives to protein language models. For instance, the recent release of AlphaGenome attempts to use DNA sequences to perform downstream tasks directly. I wonder whether Caduceus can be extended to also support DNA sequences.
>
> Response: Thank you for raising the insightful question. Proteins and genes constitute distinct and significant functional entities in life science. Caduceus leverages the MLLM paradigm to align biological sequence with natural language, thereby facilitating textual interpretation of proteins. **Thanks to the plug-and-play efficiency of the MoLE module and the scalability of the constructed instruction tuning framework, Caduceus can be readily extended to support the decoding and interpretation of DNA sequences, simply by replacing the protein encoder with a corresponding DNA sequence model such as Evo [9] and AlphaGenome [10].** Actually, this is one of the directions under active investigation in our subsequent work.
>
>
> References:
>
> [1] Uniprot: a worldwide hub of protein knowledge. Nucleic Acids Research, 47(D1):D506–D515, 2019.
>
> [2] The protein data bank. Nucleic Acids Research, 28(1):235–242, 2000.
>
> [3] The enzyme database in 2000. Nucleic Acids Research, 28(1):304–305, 2000.
>
> [4] Gene Ontology: tool for the unification of biology. Nature Genetics, 25(1):25–29, 2000.
>
> [5] Evaluating Protein Transfer Learning with TAPE. NeurIPS 2019.
>
> [6] FLIP: Benchmark tasks in fitness landscape inference for proteins. NeurIPS 2021.
>
> [7] PEER: A Comprehensive and Multi-Task Benchmark for Protein Sequence Understanding. NeurIPS 2022.
>
> [8] Structure-based protein function prediction using graph convolutional networks. Nature Communications, 12(1):1–14, 2021.
>
> [9] Sequence modeling and design from molecular to genome scale with Evo. Science, 386(6723):eado9336.
>
> [10] AlphaGenome: advancing regulatory variant effect prediction with a unified DNA sequence model. BioRxiv, pages 2025–06.

---

### Author Response · Authors · 2025-12-04
**AC Letter: Summary (2/2)**

**Rebuttal Responses:**

1. Reviewer 6ui8's primary focus lies on the performance gains when scaling the model parameters from 650M to 3B. We therefore conduct extensive new evaluation results on Caduceus-3B and Caduceus-3B-Instruct. As shown in Table 1 (Page 6, Line 300) and Table 4 (Page 8, Line 380), **parameter scaling yield clear performance gains on 24 out of 25 evaluation metrics (especially leading performance improvements of 13.2% on AAV, 16.3% on Flu, 15.73% on GO-CC, and 10.11% on BLUE-4)**. We believe this demonstrates that our model possesses strong scalability and generalizability.

2. Reviewer xRjq suggests including the performance comparison of our work with a recent highly related work Kara [1] presented at ICML 2025. Following the valuable advice, we have incorporated representative experiments from Kara [1] (*i.e.*, amino acid contact prediction and protein interaction identification) into the newly added Section 4.5. As presented in Table 5 and Table 6 (Page 9, Line 474), we are excited to report that **our model achieves new state-of-the-art performance on all 18 evaluation metrics, surpassing the previous best performance established by Kara**.

3.1. Reviewer x9cp suggests that we distinguish the main technical distinction between our work and the prior study [2]. In contrast, **we innovatively propose the property-guided gating router to assign domain-specific protein tokens to different LoRA experts**. Our MoLE module receives multi-modal inputs and assigns the protein tokens to distinct experts guided by pre-aggregated prototypes of property descriptions, thereby effectively mitigating the severe data interference issue prevalent in protein multi-modality learning.

3.2. Reviewer x9cp is also concerned about potential data leakage issues. And we clarify that **all pre-training sequences belonging to the same protein families as those in the evaluation benchmarks** (*i.e.*, with maximum amino acid similarity < 40% using the CD-HIT clustering algorithm [3]) **are thoroughly excluded to avoid data leakage** following previous works [4, 5].

4.1. Reviewer UYrt overlooks the architectural innovation of our work and defines this research as a deployment application of MoLE. To facilitate accurate understanding, we highlight that **we propose the novel property-guided routing strategy and two mulit-modal training objectives** (*i.e.*, Biotext-guided Static segment Reconstruction (BSR) and Property-grouped Dynamic segment Alignment (PDA)). The synergy of newly proposed property-guided gate routing strategy and MoE-enhanced training objectives enables discriminative encoding of protein tokens corresponding to different properties, effectively unifying biological and natural language to serve as powerful multi-modal foundation models.

4.2. Reviewer UYrt suggests that we expand the ablation experiments on MoLE architecture. Following the kind advice, **we have newly incorporated the ablation analyses of the included number and loading rank of LoRA experts** in Table 7 (Page 10, Line 496) and Figure 5 (Page 10, Line 520). Moreover, the newly added baseline in Table 4 (Page 8, Line 380) under **the identical training setting (only excluding MoLE) exhibits consistently inferior performance**, which further validates the effectiveness of our innovative architectural design.


We have endeavored to thoroughly address all reviewers' questions to facilitate precise understanding. And we have conducted extensive additional experiments, achieving new state-of-the-art performance on 18 novel evaluation metrics, supported by comprehensive ablation analyses. The manuscript has been expanded to 10 pages based on the substantial new content introduced during the rebuttal phase. We kindly invite you to consider our detailed clarifications and supplementary experiments provided during the rebuttal phase when finalizing your decision. We would also be grateful if you could take into account the dedication invested in our work and its potential contributions to future research in related fields.

Thank you again for your time and support in our submission.

Sincerely,

Authors of Paper 13526


[1] Retrieval-Augmented Language Model for Knowledge-aware Protein Encoding. In International Conference on Machine Learning, 2025.

[2] Mixture of LoRA Experts. ICLR 2024.

[3] CD-HIT: accelerated for clustering the next-generation sequencing data. Bioinformatics, 2012.

[4] ProtST: Multi-modality learning of protein sequences and biomedical texts. ICML 2023.

[5] Towards General-Purpose Protein Understanding with LLMs. SIGKDD 2025.

---

### Author Response · Authors · 2025-12-04
**AC Letter: Summary (1/2)**

Dear ACs,

We would like to express our sincere gratitude to all the area chairs and reviewers for assessing our revised manuscript and detailed responses.

We are pleased to receive overall positive comments from all reviewers. In terms of research motivation, all four reviewers (6ui8, xRjq, x9cp, UYrt) consistently highlight **the research motivation is clear and the paper is well written** with figures as visual illustrations. As for proposed method, all four reviewers (6ui8, xRjq, x9cp, UYrt) universally recognize that utilizing MoLE to unify biological and natural language is **an innovative approach grounded in the important research direction**. With respect to experimental results, three reviewers (6ui8, xRjq, x9cp) indicate **the solid empirical results** (*i.e.*, achieving state-of-the-art performance across 17 benchmarks) and detailed ablation studies adequately confirm the superiority of proposed method.


Herein, we further summarize the key contributions of our research work, along with detailed responses and revisions to address all reviewers' questions.

**Key Contributions:**

In this paper, we propose Caduceus, a family of MoE-enhanced foundation models that unify biological and natural language to unlock the textual interpretation of proteins. Our work encompasses comprehensive innovations across collected corpus, architectural design, training paradigm, and experimental performance.

a. We collect a novel large-scale, multi-modal pre-training dataset comprising over 250 million samples, and invest substantial computation resources of ~12,000 V100 GPU hours to train a family of foundation models (*i.e.*, Caduceus-650M, Caduceus-3B, Caduceus-650M-Instruct, Caduceus-3B-Instruct).

b. We propose a novel property-guided gate routing scheme within MoLE architecture, coupled with two newly customized training objectives, to mitigate the inherent data interference issue in the realm of protein multi-modality learning.

c. We develop a hierarchical training framework based on multi-modal alignment and instruction tuning, empowering large language models to comprehend biological knowledge and conduct instruction-following question-answering.

d. Caduceus remarkably achieves new state-of-the-art performance on 17 mainstream benchmarks, across 27 evaluation metrics.

---

### Meta-Review · Area_Chair_FFJk · 2026-01-09

**Summary:**

The paper proposes a Mixture-of-LORA-experts architecture to train a foundational model for protein and language data. Significant computational resources have been invested. Below is the summary of reviewer concerns:
Reviewer 6ui8:
1) QA dataset is important for model development, it is not stated if this dataset will be released or not.
2) Scaling from 650M to 3B is not clearly seen.

Reviewer xRjq:
1) No standard deviation given
2) Existing work from ICML for RAG and protein data is not mentioned.


Reviewer x9cp:
1) Limited novelty, the core MoLE mechanism has been described before, and the paper is only doing domain adaptation.

2) Train/test overlap concern, maybe there is data leakage.

Reviewer UYrt
1) Method not novel enough
2) Improvement in Table can be due to property description during pretraining (I did not get this one)
3) Weak baselines (without specification)

**Reviewer Concerns:**

Reviewer 6ui8: fully addressed concerns.
Reviewer xRjq: full addressed concerns
Reviewer x9cp: The main concern not fully addressed. Some engineering things are mentioned, but they are more domain-specific, rather that a new architecture.
Reviewer UYrt: I think addressed, since not a good review.

**Reviewer Scores:**

My guess it that the ratings will be 6, 6, 4, 4.

---

### Decision · Program_Chairs · 2026-01-26

Reject